# Unraveling the Gradient Descent Dynamics of Transformers

**Bingqing Song**\*
University of Minnesota, Twin Cities
song0409@umn.edu

**Boran Han**
Amazon Web Services
boranhan@amazon.com

**Shuai Zhang**
Amazon Web Services
shuaizs@amazon.com

**Jie Ding**
University of Minnesota, Twin Cities
dingj@umn.edu

**Mingyi Hong**
University of Minnesota, Twin Cities
mhong@umn.edu

## Abstract

While the Transformer architecture has achieved remarkable success across various domains, a thorough theoretical foundation explaining its optimization dynamics is yet to be fully developed. In this study, we aim to bridge this understanding gap by answering the following two core questions: (1) Which types of Transformer architectures allow Gradient Descent (GD) to achieve guaranteed convergence? and (2) Under what initial conditions and architectural specifics does the Transformer achieve rapid convergence during training? By analyzing the loss landscape of a single Transformer layer using Softmax and Gaussian attention kernels, our work provides concrete answers to these questions. Our findings demonstrate that, with appropriate weight initialization, GD can train a Transformer model (with either kernel type) to achieve a global optimal solution, especially when the input embedding dimension is large. Nonetheless, certain scenarios highlight potential pitfalls: training a Transformer using the Softmax attention kernel may sometimes lead to suboptimal local solutions. In contrast, the Gaussian attention kernel exhibits a much favorable behavior. Our empirical study further validate the theoretical findings.

## 1 Introduction

Transformer model architectures have become popular in machine learning, delivering remarkable performance across a wide array of tasks. From natural language processing [Vaswani et al., 2017, Beltagy et al., 2020] to computer vision [Dosovitskiy et al., 2020], these models have set new standards in performance and efficiency. Popular models include BERT [Devlin et al., 2018], RoBERTa [Liu et al., 2019], DeBERTa [He et al., 2020], GPT models [Radford et al., 2019, Brown et al., 2020] and ViT [Dosovitskiy et al., 2020]. Despite their empirical success, a comprehensive understanding of their optimization process remains elusive. As highlighted in Liu et al. [2020], the training of large Transformers can sometimes result in deteriorated performance. It is therefore critical to develop theoretical insights for researchers and practitioners to better understand the practical performance of Transformers. However, the complexity of their architectures, coupled with the non-convex nature

---

\*The work of B. Song was partially done while interning at Amazon Web Services.

38th Conference on Neural Information Processing Systems (NeurIPS 2024).

of the associated optimization problems, has made the theoretical analysis of these models very challenging.

The optimization landscape can be pivotal for understanding a certain type of neural network and providing the practical guidance [Liu et al., 2020]. Existing literature offers numerous studies on achieving zero-loss solutions in networks with ReLU activation. These studies encompass various network structures, including fully-connected, convolutional, and residual networks, as explored in [Jain et al., 2017], [Jin et al., 2021], and [Danilova et al., 2022]. They delve into the analysis of network optimization landscapes and provide assurances of rapid global convergence when using gradient descent (GD) or stochastic gradient descent (SGD) algorithms. For instance, in Du et al. [2019], the authors focus on fully-connected networks and ResNets with smooth activation functions, and they have demonstrated that global convergence can be achieved using GD with a network size proportional to $\mathcal{O}\big(\text{poly}(N)\big)$, where $N$ is the sample size. Similarly, [Allen-Zhu et al., 2019] show that ReLU fully-connected networks with at least $\mathcal{O}\big(\text{poly}(N)\big)$ neurons can achieve global convergence using GD or SGD. From a statistical perspective, [Li et al., 2023] have shown that for two-layer ReLU neural networks (with input dimension $p$) that admit a sparse subnetwork representation, a sample size of $O(\log^4(p/\delta))$ can guarantee the global convergence with probability at least $\delta$ using GD. Despite this extensive body of work on traditional architectures, it is not clear what conditions we need (e.g. network size, optimizer, initialization) to ensure training Transformer models to find high-quality solutions.

Compared to traditional deep learning architectures, Transformers incorporate a unique level of intricacy through their attention kernel [Vaswani et al., 2017], which is designed to effectively handle sequence inputs. This mechanism incorporates Softmax activation to the inner products of query and key vectors, and this inherently non-convex operation poses considerable challenges to theoretical analysis. Consequently, existing frameworks for analyzing the convergence of classical deep learning models are not directly applicable to Transformers. Further, many recent works have pointed out that the performance of Transformers depends on a number of factors such as the choice of kernel function, initialization, choice of optimizers, and forms of token embeddings [Huang et al., 2020, Pan and Li, 2023, Shazeer, 2020, Li et al., 2018, Tian et al., 2023]. In deep learning, these factors have been studied in a line works. For example, Li et al. [2018] show that the good training performance is not universal ; skip connections have the effect of smoothing the training landscape, and the Adam algorithm tends to follow a more direct trajectory towards optimal solutions compared to SGD. Therefore, it is imperative to understand what kind of conditions, including initialization, network structure, data properties, and optimizer choices, will lead to high-performing Transformers.

In this work, we will delve into the intricacies of attention kernels, discussing both their advantages and limitations in the context of model optimization. The main contributions of this work are threefold.

- We derive the conditions that will make the one-layer Softmax attention Transformer reach global optimality with vanilla gradient descent. The convergence guarantee is largely attributed to the linear layer ($W^V$) in the attention mechanism.
- We investigate the attention kernel's effectiveness, revealing Gaussian attention achieves zero training loss, while Softmax can lead to non-optimal stationary points.
- Our experiments validate that Softmax attention Transformers converge slower and present more challenging training landscapes than Gaussian counterparts, potentially leading to more local optimal solutions.

## 2 Related Work

A number of research works have focused on the theoretical analysis and interpretation of Transformer models, revealing crucial insights into their practical performance.

Liu et al. [2020] showed that heavy reliance on the residual branch in multi-layer Transformer models can lead to training instability, which amplifies small parameter perturbations, causing significant disturbances in the model's output. In Bhojanapalli et al. [2020], the authors illustrated the existence of a low-rank bottleneck in Transformer models with sufficiently large embedding and hidden size ($D = d$). However, this work focuses on the representation ability of large size attention, while falling short of analyzing Transformer models from an optimization perspective. In Noci et al. [2022],

the authors explored rank collapse issues in token representations and their impact on training. The authors discussed the origin of the phenomenon of rank collapse and proposed depth-dependent scaling of residual branches as a potential solution. They specifically investigated scenarios where token rank equals one, which can hinder Transformer training. Their findings demonstrate the occurrence of the vanishing gradient issue, however, this work does not comprehensively characterize the vanishing gradient problem throughout the entire training process.

A recent work Wu et al. [2024] analyzes the convergence behavior of shallow Transformer, which builds a convergence theory of shallow Transformer with realistic structure and initialization, but they do not provide roles for different matrices in the convergence. However, the focus of our paper is different from Wu et al. [2024]. We not only derive the global convergence analysis (Our Theorem 2), but also investigates the role of different variables in optimization.

Some other works focus on improving the optimization of Transformers empirically. [Huang et al., 2020] have proposed an initialization strategy such that no warm-up or layer normalization is needed to train Transformers efficiently; in Shazeer [2020], the GLU variant of token embedding has been showed to be better than plain embedding in the optimization of Transformer models with Softmax attention kernel. It is worth noting that the above works all primarily focus on empirical investigations into the training of Transformer models, lacking a comprehensive theoretical analysis of the underlying mechanisms.

Some recent research has focused on the convergence analysis of Transformer-based models within the in-context learning (ICL) framework. For instance, Huang et al. [2023], Zhang et al. [2023] explores the learning dynamics of a one-layer Transformer with Softmax attention trained via gradient descent to learn linear function classes in-context. However, this line of study primarily addresses the general convergence performance of Transformers within the ICL setting and does not delve into the role of individual variables. More specifically, these works analyze the convergence of in-context training, where a prompt is constructed with all the training samples and a single test sample. The goal of these works is to achieve the zero test loss (in expectation) by optimizing over the loss function modeled by the prompt. On the other hand, our analysis is based on standard empirical loss minimization, which does not involve any prompt construction.

## 3 Notations and Problem Description

In this section, we define the structure of the Transformer model and describe the training problem. We consider a one-layer attention Transformer model with multiple heads and a dataset with $N$ samples. Each data sample consists of $n$ discrete tokens, each with embedding dimension $D$. We denote the dataset as $\{(X_i, y_i)\}_{i=1}^N$, where $X_i \in \mathbb{R}^{n \times D}$, and $y_i \in \mathbb{R}^n$ is the label of the dataset. The output from the Transformer model is the prediction of the label. The Transformer structure is formulated as follows:

$$\text{Attention}(W_h^Q, W_h^K, W_h^V; X_i) := S(W_h^Q, W_h^K; X_i) X_i W_h^V \tag{1}$$

$$\text{MH}(W^Q, W^K, W^V; X_i) := \text{Concat}(\text{head}_1, \ldots, \text{head}_H) \cdot W^O,$$

$$\text{where } \text{head}_h := \text{Attention}(W_h^Q, W_h^K, W_h^V; X_i), h = 1, \cdots, H. \tag{2}$$

In the above notation, $W_h^Q, W_h^K \in \mathbb{R}^{D \times d}$ is the query weight matrix and key weight matrix, respectively; $W_h^V \in \mathbb{R}^{D \times d}$ is the value weight matrix; these matrices are the main optimization variables throughout the paper. Further $W^O \in \mathbb{R}^{Hd \times 1}$ is a fixed matrix, representing the weight of the output layer; $H$ is the number of attention heads; $S(\cdot)$ is a kernel function of variables $W^Q, W^K$ and input $X_i$. Attention$(\cdot)$ is the attention head function; MH$(\cdot)$ represents the multi-head attention function. For example, with the Softmax attention [Vaswani et al., 2017], $S(\cdot)$ can be written as:

$$S\left(W_h^Q, W_h^K; X_i\right) := \text{Softmax}\left(\frac{X_i W_h^Q \left(X_i W_h^K\right)^\top}{\sqrt{d}}\right) \tag{3}$$

where for a given $n \times n$ matrix $Z$, $\text{Sofmax}(Z) := [\text{Softmax}(Z_1), \cdots, \text{Softmax}(Z_n)]$. Throughout, let us denote $S(\cdot)_{kj}$ as the element of $k$-th row and $j$-th column in matrix $S(\cdot)$. Let $X_{ik.} \in \mathbb{R}^D$ denote the embedding of the $k$-th token in data $X_i$, which is the $k$-th row of matrix $X_i$. The structure of Transformer model can be found in Fig 1, where we denote $S_{ih} := S\left(W_h^Q, W_h^K; X_i\right)$.

Based on the above Transformer model, we consider minimizing the following empirical $\ell_2$ loss function for the entire data set $\{X_i, y_i\}_{i=1}^N$:

$$\min_M \frac{1}{2} \sum_{i=1}^N \|\mathsf{MH}(M; X_i) - y_i\|^2, \tag{4}$$

where $M := (W^Q, W^K, W^V)$ is the set of variables that can be optimized.

For notation simplicity, next we define the vector version of the Transformer model given in Equation (1), for the entire dataset $\{(X_i, y_i)\}_{i=1}^N$. Towards this end, let $X \in \mathbb{R}^{Nn \times D}$ denote the column-stacked matrix of each single data $X_i$. Similarly, define the stacked label $y \in \mathbb{R}^{Nn}$. Then we can define:

$$\mathsf{MH}(M; X) := \begin{pmatrix} S_{11}X_1 & \cdots & S_{1H}X_1 \\ \cdots & \cdots & \cdots \\ S_{N1}X_N & \cdots & S_{NH}X_N \end{pmatrix} \cdot \mathrm{diag}(W_1^V, \cdots, W_H^V) \cdot W^O, \tag{5}$$

$i = 1, 2, \cdots, N, h = 1, 2, \cdots, H$ for simplicity.

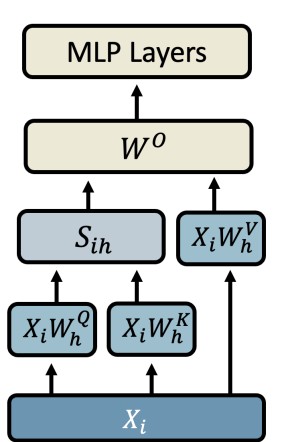

MLP Layers

$W^O$

$S_{ih}$  $X_i W_h^V$

$X_i W_h^Q$  $X_i W_h^K$

$X_i$

Figure 1: One head in Transformer architecture with Softmax Attention.

Let $B := \begin{pmatrix} S_{11}X_1 & \cdots & S_{1H}X_1 \\ \cdots & \cdots & \cdots \\ S_{N1}X_N & \cdots & S_{NH}X_N \end{pmatrix}$, and $W^V :=$ $\mathrm{diag}(W_1^V, \cdots, W_H^V) \in \mathbb{R}^{HD \times Hd}$ denote the diagonalized weight matrices that include all value weight matrices for all attention heads. Using these definitions, We can simplify Equation (5) as

$$\mathsf{MH}(M; X) = B \cdot W^V \cdot W^O$$

Thus the empirical loss function given in Equation (4) can be simplified as

$$\min_M \frac{1}{2} \|\mathsf{MH}(M; X) - y\|^2. \tag{6}$$

For more notations in the following sections, we will use subscript $t$ to represent the variables in $t$-th iteration, e.g, $M_t := \{W_t^Q, W_t^K, W_t^V\}$. Similarly, we denote $B_t$ as the matrix $B$ at $t$-th iteration.

It is important to note that, in the above description and throughout the paper, we model the Transformer training problem by using a single-layer Transformer, with a regression loss. In practice Transformer models can exhibit greater complexity (different loss functions, multiple layers, etc). For example, the text classification task has an additional mean pooling layer followed by the output of the Transformer structure. Further, they usually contain downstream MLP modules. However, we choose to use the simplified version due to the following reasons:

First, the primary objective of this work is to understand how different attention kernels affect the training dynamics of the Transformers, so we do not include the layer normalization in our model. In fact, in the literature, many works that analyze popular network structures also do not consider layer normalization. For example, in [Huang et al., 2023, Zhang et al., 2023], both analyze the convergence performance of Transformers but normalization is not considered.

Second, we do not include the downstream MLP module in our work since we are interested in the role of self-attention layer in convergence analysis, and the single-attention model is also the standard model used in [Huang et al., 2023, Zhang et al., 2023]. Further, the analysis of MLP is standard in literature [Allen-Zhu et al., 2019, Du et al., 2019, Nguyen and Mondelli, 2020]. And it is worth noting that our choice to focus on a one-layer Transformer is consistent with other works that similarly aim to investigate the core training dynamics of Transformers, e.g, in [Tian et al., 2023], a single-layer Transformer is considered as a basic model.

# 4 Convergence Analysis

In this section, we present our theoretical analysis for solving problem (6). We focus on the behavior of the vanilla GD algorithm for optimizing the variable set $M$, where $M \subset \{W^Q, W^K, W^V\}$. Below we summarize our results.

**Common convergence conditions with Softmax Attention**: When the activation function $S(\cdot)$ is either the Softmax or Gaussian function, and the embedding dimension $D$ is at least $\mathcal{O}(Nn)$, optimizing Equation (6) can achieve a global optimal solution when $M = \{W^V\}$ and $M = \{W^Q, W^K, W^V\}$.

**Different behavior between Softmax and Gaussian Kernel Attention**. When $S(\cdot)$ is Gaussian and the embedding dimension $D$ is at least $\mathcal{O}(Nn)$, convergence to global optimal is also ensured for $M = \{W^Q\}$. Interestingly, under the same conditions of large $D$, convergence to global optimal is *not* guaranteed when $S(\cdot)$ is Softmax.

In the subsequent sections, we will elaborate on these convergence results in detail, providing a deeper understanding of the nuances in Transformer behavior under varying configurations. To set up our analysis, we introduce $\underline{\lambda}^V$ as the smallest eigenvalue of $W_0^V$, $\underline{\lambda}^B$ as the smallest eigenvalue of $B_0$, $\bar{\lambda}_h^Q, \bar{\lambda}_h^K, \bar{\lambda}^V$ as the largest singular value of matrix $W_{h,0}^Q, W_{h,0}^K, W^V$, respectively. We denote $\| \cdot \|_2$ as $\ell_2$ norm and $\| \cdot \|_F$ as Frobenius norm. Further, we denote $\sigma_{\max}(\cdot)$ and $\sigma_{\min}(\cdot)$ as the largest and smallest singular value of a matrix, respectively. For any vector $v$, let $\min(|v|)$ denote the smallest absolute value of vector $v$.

## 4.1 Convergence to global optimal

First, we examine the role of $W^V$ in the optimization of multi-head attention network structure. Our analysis demonstrates that with the hidden dimension $HD \geq Nn$ and proper initialization, the global optimal solution of (6) can be found using a vanilla gradient descent algorithm. The initialization requires that the matrix $B_0$ has full rank. Our first result shows that, overparameterized Transformer can be trained to global optimal solution.

**Theorem 1.** *Consider problem (4) with $S(\cdot)$ being instantiated as the Softmax kernel given in (3). Consider the following update for the variable $M = \{W^V\}$: $W_{t+1}^V = W_t^V - \eta \nabla_{W^V} f(M_t; X)$, where $\eta > 0$ is the stepsize.*

*Suppose $W_0^Q$ and $W_0^K$ are initialized such that $\underline{\lambda}^B > 0$. Then we have:*

$$f(M_t; X) \leq (1 - \eta\alpha)^t f(M_0; X), \tag{7}$$

*where $\alpha := \|W^O\|^2 (\underline{\lambda}^B)^2 > 0$; $\eta > 0$ is defined in Appendix 1.3, and chosen such as $\eta\alpha < 1$.*

**Remark 1.** *The aforementioned theorem focuses on the convergence behavior when only $W^V$ is being updated. We further elaborate on the initial conditions ensuring $\underline{\lambda}^B > 0$.*

*Note that $\underline{\lambda}^B > 0$ implies that the objective function $f$ exhibits a landscape that is nearly convex, which is crucial for optimization. By definition, this condition implies that $B_0$ has full rank, which can be fulfilled by selecting appropriate $W_0^Q$ and $W_0^K$, plus having large enough embedding size, satisfying $D \geq Nn/H$. We refer the readers to Appendix 1.3 for the derivation of this condition, which can be guaranteed by random initialization with high probability.*

Furthermore, it is important to note that our work aligns with existing literature on the subject of embedding size in Transformer models. For example, in [Bhojanapalli et al., 2020], the authors restrict their focus to the simplified case of $N = 1, H = 1$. They establish the necessary condition for Softmax attention to overcome its low-rank bottleneck, which requires $D \geq n$. In our analysis, we derive a similar necessary condition on Transformer model size ($D \geq n \times (N/H)$) to guarantee the global convergence when a Transformer model is trained with GD.

In Theorem 1, we have illustrated the case where only updating $W^V$ already leads to global convergence. However, in practice, all parameters $W^V, W^Q, W^K$ are updated. This case is more challenging to analyze due to the non-linearity introduced by the Softmax function. Next, we show that a similar result in Theorem 1 still holds when all the parameters are updated simultaneously.

**Theorem 2.** *Consider problem (4), with $S(\cdot)$ being instantiated as the Softmax kernel. Consider the GD update where $M = \{W^Q, W^K, W^V\}$: Suppose $\underline{\lambda}^B > 0$, and the initialization $M_0$ satisfy*

$$\frac{n^2\sqrt{NH}\|X\|_F^5 \sum_{h=1}^{H} \left((\bar{\lambda}_h^Q)^2 + (\bar{\lambda}_h^K)^2\right) \bar{\lambda}^V}{\|W^O\|_2 \cdot (\underline{\lambda}^B)^2 \min(\bar{\lambda}_h^Q, \bar{\lambda}_h^K, \underline{\lambda}^B)} \times \|\mathsf{MH}(M_0; X) - y\|_2 \leq \nu. \tag{8}$$

*Then there exists stepsize $\eta > 0$, such that*

$$f(M_t; X) \leq (1 - \eta\beta)^t f(M_0; X), \tag{9}$$

*where $\beta := \|W^O\|^2 (\underline{\lambda}^B)^2 > 0$, and the constants $\eta, \nu$ are defined in Appendix 1.3.*

**Remark 2.** *In the stated theorem, we simplify our analysis by excluding the downstream MLP module in the typical Transformer model, since it is easy to combine the model in Equation (2) with downstream MLP layers. Further, it can be directly showed that the Transformer with MLP will lead to the **same** convergence rate of the optimization problem as updating $W^Q, W^K, W^V$ only. To illustrate this, consider the following Transformer model:*

$$G\left(W^Q, W^K, W^V; X_i\right) = \mathsf{MH}(W^Q, W^K, W^V; X_i) \cdot W^1 W^2 \cdots W^L, \tag{10}$$

*where $W^l \in \mathbb{R}^{n_{l-1} \times n_l}$, and $n_0 = d^O$. Based on the Transformer model defined in Equation (10), we have the following corollary.*

**Corollary 1.** *Consider problem $\min_M \frac{1}{2}\|G(M; X) - y\|^2$, with $G(\cdot)$ being defined in Equation (10) and $S(\cdot)$ being instantiated as the Softmax kernel. Suppose that the MLP module satisfies:*

$$n_1 \geq n_2 \cdots \geq n_L.$$

*Consider the following GD update (where $M = \{W^Q, W^K, W^V, W^1, \cdots, W^L\}$): Suppose $\underline{\lambda}^B > 0$. Then, there exists a step size $\eta > 0$ and initialization weight $M_0$, such that the loss function linearly converges to $0$.*

**Remark 3.** *The above theorem and corollary describe the global convergence guarantee when $W^Q, W^K$ and $W^V$ are updated. This is in line with the insights gained from Theorem 1. However, the conditions for initialization are more stringent, and the optimization landscape becomes inherently more complex due to the involvement of the Softmax attention through $W^Q$ and $W^K$.*

*To ensure the initial condition 8, we have two options: 1) Initializing $M_0$ such that $\|\mathsf{MH}(M_0; X) - y\|_F$ is small, which implies that the optimization starts in a region close to the global optimal solution and that the initial weight is close to the global optimal solution; 2) Balancing between $W^O$ and $W^V$, in the sense that $\|W^O\|_2$ is large and $\bar{\lambda}^V$ is small. For a detailed account of these initialization strategies, please refer to Appendix 1.3.*

*Finally, we need to point out that for Transformers with Gaussian kernel attention, we can derive similar convergence results as long as the attention kernel maintains full rank and weights are initialized appropriately. Here we do not include the theoretical statement since it is similar to the result for Softmax attention.*

## 4.2 Softmax vs Gaussian kernel: Softmax attention Transformers may exhibit slower convergence.

In the previous section, we explored the global convergence of training Transformer models. However, from Theorem 2, it was not clear what roles do matrices $W^Q$ and $W^K$ play in the entire convergence process, since Theorem 1 indicates that optimizing $W^V$ alone already ensures the desired convergence. Nevertheless, it is the matrices $W^K$ and $W^Q$ that truly represent the power of a Transformer model, because they are used to extract token correlations.

To study how well a Transformer model can extract the token correlation, in this section, we will study the GD dynamics for Transformer models, where only $W^K$ and $W^Q$ are optimized (while fixing $W^V$). If optimizing these two parameters alone can still achieve zero training loss, then we claim that the input token correlation can be optimally extracted by the Transformer model.

#### 4.2.1 Notations

To begin our study, let us define that Gaussian kernel to be an $n \times n$ matrix, where its $k$-th row and $j$-th column of is given by:

$$S\left(W_h^Q, W_h^K; X_i\right)_{kj} = \exp\left(-\frac{1}{\sqrt{d}}\left(X_{ik\cdot}.W_h^Q - X_{ij\cdot}.W_h^K\right)^2\right) \tag{11}$$

Since the training dynamics/gradients of variables $W^Q$ and $W^K$ have the same property in (3) and (11), we will only concentrate on optimizing $W^Q$.

With some abuse of notation, define a matrix $C$ for Softmax attention and Gaussian kernel attention, respectively. Softmax attention: $C_{ih} := \frac{X_i W_h^Q \left(X_i W_h^K\right)^\top}{\sqrt{d}} \in \mathbb{R}^{n \times n}$.

Gaussian kernel attention: $C_{ih} \in \mathbb{R}^{n \times n}$; $(C_{ih})_{kj} = -\frac{\left\|X_{ik\cdot}.W_h^Q - X_{ij\cdot}.W_h^K\right\|^2}{2\sqrt{d}}$.

For both Softmax attention and Gaussian kernel attention:

$$C_i \in \mathbb{R}^{n \times Hn} = [C_{i1}, C_{i2}, \cdots, C_{iH}]; \; C \in \mathbb{R}^{Nn \times Hn} = \left[C_1^\top, C_2^\top, \cdots, C_N^\top\right]^\top.$$

Using the above notation, the activation function $S(\cdot)$ in (3) and (11) can be related to the matrices $C$'s in the following manner:

$$\text{Softmax attention}: S_{ih} = \text{Softmax}\left(C_{ih}\right), \; \text{Gaussian attention}: (S_{ih})_{kj} = \exp\left((C_{ih})_{kj}\right).$$

Additionally, note that $C$ is a function of variables $M$. Therefore we will sometimes use $C(M)$ when we need to emphasize the dependency of $C$ on $M$.

#### 4.2.2 Main Results

Next, we will outline the conditions under which GD can still successfully find global optimal solutions for Transformers with Gaussian kernel attention (when only $W^Q$ is updated), while under the same set of conditions, but with Softmax kernel attention, GD fails.

**Theorem 3.** *Solve problem (4) with the following GD update (with $M = \{W^Q\}$): $W_{t+1}^Q = W_t^Q - \eta \nabla_{W^Q} f(M_t; X)$. Suppose $\delta_h := \sigma_{\min}(\frac{\partial C(M_0)}{\partial W_h^Q}) > 0, \; \forall \; h \in [1, 2, \cdots, H]$, and the initialization condition further satisfies*

$$\frac{n\|X\|_F^5\left(\bar{\lambda}_h^Q + \bar{\lambda}_h^K\right)\exp\left(\frac{9}{4}\|X\|_F^2\left((\bar{\lambda}_h^Q)^2 + (\bar{\lambda}_h^K)^2\right)\right)}{\left(\min(|V'W^O|)\right)^2 \cdot \min(\delta_h, \bar{\lambda}_h^Q)} \times \bar{\lambda}^V\|W^O\|_2 \cdot \|\mathsf{MH}\left(M_0; X\right) - y\|_2 \leq \nu', \tag{12}$$

*$\nu'$ is defined in Appendix 1.5.*
*(1) When $S(\cdot)$ is a Gaussian kernel function, there exists a stepsize $\eta$ and a positive constant $\gamma$, such that*

$$f\left(M_t; X\right) \leq (1 - \eta\gamma)^t f\left(M_0; X\right), \tag{13}$$

*where $\gamma, \eta$ are defined in Appendix 1.5.*
*(2) When $S(\cdot)$ is a Softmax function, suppose $W_t^Q$ is bounded during the training phase, then there exists stepsize $\eta$, such that*

$$f\left(M_t; X\right) \leq f\left(M_0; X\right) - \eta'\sum_{r=0}^{t-1}\|\nabla_{W^Q} f\left(M_r; X\right)\|^2, \tag{14}$$

*where $\eta'$ is defined in Appendix 1.5.*

**Remark 4.** *First, it's important to note that the parameter size must satisfy $Dd \geq Nn^2$ for $\delta > 0$ to hold. It is crucial to emphasize the fundamental distinction in convergence outcomes between Transformers employing Gaussian kernel attention and those utilizing Softmax attention under these conditions. With equivalent initialization conditions, training Transformers equipped with Gaussian kernel attention achieves global convergence using gradient descent (GD). Second, it is essential to emphasize that the dimension size $Dd \geq Nn^2$ is similar to the findings of works that have analyzed the convergence performance of over-parameterized neural networks Allen-Zhu et al. [2019], Du et al. [2019]. The total number of samples, consisting of $N$ samples each with $n$ tokens, can be calculated as $Nn$. Meanwhile, the total feature dimension is $Dd$. The inequality implies that the width of the*

*parameters is at least $\mathcal{O}(N)$, a relationship also illustrated in Nguyen and Mondelli [2020]. The proof consists of two basic steps. The first step is to derive the closed form gradient of loss function over variable $W^Q$. Intuitively, the gradient of Softmax attention is much more complicated than the Gaussian attention Transformer, which will lead to a more complicated landscape and more local solutions. The second step is to analyze the gradient of Transformers with both kernels and the same initialization Equation (12). For Gaussian attention Transformer, it can be iteratively shown during the gradient descent training: 1) The variable $W^Q$ is bounded; 2) The PL condition holds (i.e, the optimization landscape remains near-convex); 3) The loss function decreases linearly. For the Softmax attention Transformer, there is no guarantee that the PL condition holds during iterative gradient descent update.*

In part (2), we demonstrate that the PL condition does not hold. In particular, we identify an initial solution that satisfies all the conditions given in Theorem 3, yet fails to satisfy the PL condition. Therefore, in this case, GD leads to vanishing gradients without being able to find a global optimal solution. The details of this specific example are provided below.

**Example:** Consider Transformer with Softmax attention, and $N = 1, n = 2, H = 1$. Let us first write down the close form of the gradient over $W_1^Q$:

$$\frac{\partial f\left(M_0; X_1\right)}{\partial W_1^Q} = \frac{1}{\sqrt{d}} X_1^\top \frac{\partial f\left(M_0; X_1\right)}{\partial C_{11}} X_1 W_{1,0}^K$$

Next, we show there exists $W^O, W^V, X_1, W_{1,0}^Q, W_{1,0}^K$ such that the loss function is non-zero with Equation (12) satisfied, while

$$\frac{\partial f\left(M_0; X_1\right)}{\partial C_{11}} = \mathbf{0} \in \mathbb{R}^{2 \times 2}.$$

Denote $L := \frac{\partial f(M_0; X_1)}{\partial \mathsf{MH}(M_0; X_1)} \left(W^O\right)^\top \left(X_1 W_0^V\right)^\top \in \mathbb{R}^{2 \times 2}$. $\frac{\partial f(M_0; X_1)}{\partial C_{11}}$ can be expressed as follows:/

$$\left(\frac{\partial f\left(M_0; X_1\right)}{\partial C_{11}}\right)_{11} = \delta \cdot (L_{11} - L_{12}), \quad \left(\frac{\partial f\left(M_0; X_1\right)}{\partial C_{11}}\right)_{12} = \delta \cdot (L_{12} - L_{11}), \delta \text{ is some constant.}$$

Next, we will give the value of $W^O, W_0^V$ to show the case where GD leads to vanishing gradient. Let $D = d = 2, W^O = (\frac{1}{a}, \frac{1}{a}), X_1 = \begin{pmatrix} 1 & 0 \\ 0 & 1 \end{pmatrix}$, and $W_0^V = \begin{pmatrix} 2a & a \\ a & 2a \end{pmatrix}$, where $a$ is a constant. It is easy to show that there exists $W_1^Q$ and $W_1^K$ such that Equation (12) holds. Further, it is easy to verify that for this scenario, the following holds:

$$L_{11} = L_{12}, L_{21} = L_{22}. \tag{15}$$

Next, we can easily deduce that $\left(\frac{\partial f(M_0; X_1)}{\partial C_{11}}\right)_{11} = \left(\frac{\partial f(M_0; X_1)}{\partial C_{11}}\right)_{12} = 0$. Similarly, we can demonstrate that $\left(\frac{\partial f(M_0; X_1)}{\partial C_{11}}\right)_{21} = \left(\frac{\partial f(M_0; X_1)}{\partial C_{11}}\right)_{22} = 0$. Consequently, we have $\frac{\partial f(M_0; X_1)}{\partial W_1^Q} = \mathbf{0}$. However, if $y_1$ satisfies that $\frac{\partial f(M_0; X_1)}{\partial \mathsf{MH}(M_0; X_1)} \neq \mathbf{0}$, it follows $f(M_0; X_1) \neq 0$, which means $M_0$ is not global optimal solution.

## 5 Experiment: Softmax v.s. Gaussian

In this section, we present numerical results to illustrate the behaviors of Transformers models with Softmax attention and Gaussian kernel attention across various tasks.

### 5.1 Dataset

We investigate two distinct tasks: Text Classification using the IMDb review dataset [Maas et al., 2011] and Pathfinder [Linsley et al., 2018]. While both tasks involve processing

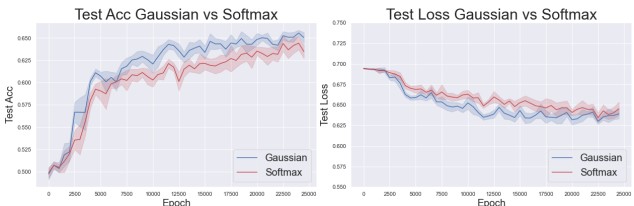

Figure 2: Test performance on text classification task with different attention kernels

long sequences, they exhibit different characteristics. Text Classification is a well-known NLP task that focuses on discerning relationships among token embeddings, while the Pathfinder task prioritizes capturing spatial information within the input pixels.

## 5.2 Model and Experiment Method

We follow the experiment setting in [Chen et al., 2021]. For both tasks, we employ a 2-layer Transformer model with the following specifications: embedding dimension $D = 64$, hidden dimension $d = 128$, and number of attention heads $H = 2$. To align the model with the classification task, we use an additional mean pooling layer as the final layer. We determine the batch size based on available memory constraints. Specifically, we set a batch size of 16 for the Text Classification task with a learning rate of $1 \times 10^{-4}$, and a batch size of 128 for the Pathfinder task with a learning rate of $2 \times 10^{-4}$. For optimization, we use Stochastic Gradient Descent (SGD) for the Text Classification task and Adam for the Pathfinder task. We conduct two types of experiments.

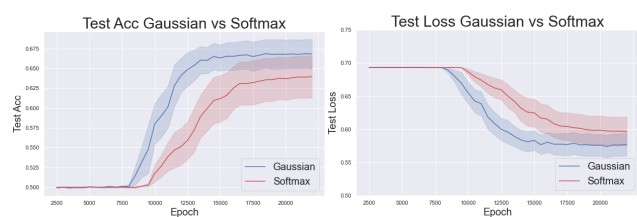

Figure 3: Test performance on pathfinder task with different attention kernels

In the first experiment, we plot the test accuracy and test loss within the training steps with both Softmax and Gaussian kernel attention on both tasks. We repeat the training for 10 times and make the shadow plot on the test performance.

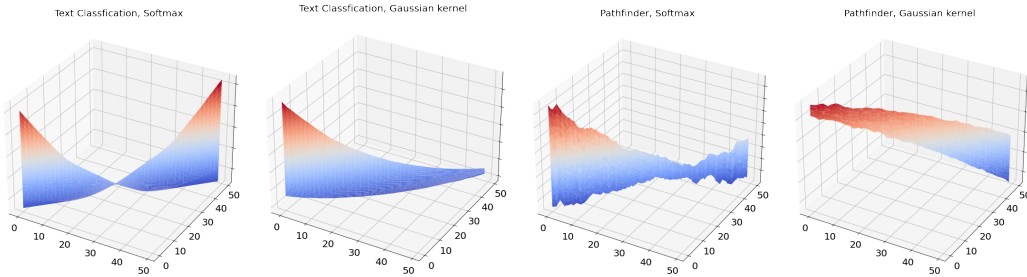

Figure 4: The loss landscapes on text classification task and Pathfinder task. For both tasks, we use the two-stage training in Section 5.2 with the same training hyperparameters, while the only difference is the attention structure in the second training stage. The two axes represent the two directions $d_1$ and $d_2$ as defined in Section 5.2.

In our second experiment, the training process consists of two stages: In the first stage, we train the Transformer model equipped with Softmax attention (defined in Equation (3)) for 8,000 steps. In the second stage, we continue training from the pre-trained model for an additional 500 steps, with the option of using either Softmax or Gaussian kernel. To explore the optimization landscape around the trained model, we employed a technique inspired by Li et al. [2018]. We select two parameter directions, specifically the $W^Q$ and $W^K$ matrices in the first Transformer layer. These two directions, denoted as $d_1, d_2$, are centered at the trained model $M$, and represent the parameter space of $W^Q, W^K$, respectively. We evaluate the loss function on the set $\{M + 0.02(r - 25)d_1 + 0.02(s - 25)d_2\}$, where $r, s \in [1, 2, \cdots, 50]$. The above set is the neighborhood of the trained model $M$, and we chose the evaluation stepsize as 0.02 along the two directions $d_1, d_2$, with the total steps limit as 100. Within this parameter space, we plot a 3-D surface representing the landscape around the trained model.

## 5.3 Results

### 5.3.1 Test Loss & Accuracy Curve comparison

To begin with, we present some observations in our first experiment. We plot the test performance of these two tasks on Transformers with two different types of attention. From Fig 2 and Fig 3, we can conclude that in both tasks, Transformers with Gaussian kernel attention exhibit faster convergence and higher test accuracy than Softmax attention with the same model size and learning rate. Especially,

training Transformers with Softmax attention in the Pathfinder task can lead to unstable performance as indicated in Fig 3. The test accuracy has a significantly higher variance at the same training epoch. Further, the worst test accuracy after $20,000$ epochs is around $0.58$ for the Softmax attention Transformer, compared with $0.62$ for the Gaussian kernel Transformer. These observations align with the experiment results in [Chen et al., 2021] and [Tay et al., 2020], where Transformers with different attention kernels are trained with the same model size and learning rate, while Softmax attention Transformers show instability in a few tasks.

### 5.3.2 Optimization Landscape Comparison

In Figure 4, we present a comparison of the optimization landscape between Transformers with Softmax and Gaussian kernel attention. Notably, we observe distinct differences in the training landscapes of these two attention types for both tasks. We follow the visualization method described in Section 5.2. We conduct a visualization of the optimization landscape around the trained models after a two-stage training process, with identical learning rates, network sizes, and training epochs. Keeping all other factors consistent, the disparity in the landscape provides a direct representation of the difference in the attention structure during the optimization procedure. With Softmax attention, the landscape appears more complicated compared with Gaussian kernel attention. This complexity can be interpreted as the presence of a greater number of local optima in the optimization landscape, suggesting that Transformers utilizing Softmax attention may encounter more challenges in reaching global optimal solutions. In contrast, the landscape with the Gaussian kernel is flatter. This observation aligns with our earlier findings in Figure 2 and Figure 3, where Softmax attention exhibited certain convergence issues. These observations also provide empirical evidence supporting our Theorem 3, which reflects in a slightly different perspective the complicated optimization landscape within the Softmax kernel.

## 6 Conclusion and Future Work

In conclusion, our study addresses critical gaps in our understanding of why Transformer models perform exceptionally well in a variety of machine learning tasks. Our work also provides a nuanced understanding of the advantages and disadvantages of using classical Softmax attention in Transformers. We find that while shallow Softmax attention Transformers can achieve global convergence with overparameterization, there are scenarios where this attention structure can lead to local solutions. However, those issues can be mitigated by the Gaussian kernel-based attention. In our work, we need strong initialization and large embedding size, i.e, $HD \geq Nn$ to obtain the global convergence, which exhibits a gap towards real case. In the future work, we will investigate how to relax the assumptions.

## 7 Acknowledgment

The work of B. Song was partially done while interning at Amazon Web Services. M. Hong holds concurrent appointments as an Amazon Scholar and as a faculty at the University of Minnesota. This paper describes their work performed at Amazon. The work of Jie Ding was supported in part by the Army Research Office Early Career Program Award under grant number W911NF2310315.

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

# 1 Appendix

## 1.1 Notations

Recall that we have defined the structure of a single Transformer model in Equation (1) and Equation (5). We will further define a few notations before we introduce a few useful lemmas that are needed in our proof.

(1) Operator: Denote $\text{vec}(\cdot)$ as the vectorization operator on a matrix; $\otimes$ as Kronecker product operator; $\odot$ as the element product. Denote $\Upsilon(\cdot)$ as a matrix operator, such that for any matrix $X$ without zero element

$$\Upsilon\left(X_{m \times n}\right) = \begin{bmatrix} 1/x_{11} & \cdots & 1/x_{1n} \\ \vdots & \ddots & \vdots \\ 1/x_{m1} & \cdots & 1/x_{mn} \end{bmatrix}_{m \times n} \tag{16}$$

(2) Matrix: Denote $\mathbb{I}$ as the identity matrix. Define matrix $\mathbb{E}$ and $E$ as following:

$$\mathbb{E} = \begin{bmatrix} E & & \\ & \ddots & \\ & & E \end{bmatrix}_{Hn \times Hn} , \quad E = \begin{bmatrix} 1 & \cdots & 1 \\ \vdots & \ddots & \vdots \\ 1 & \cdots & 1 \end{bmatrix}_{n \times n} .$$

Define matrix $\mathbb{P}_h$ as following: $\mathbb{P}_h = \left(\ldots, E_{n \times n}^h, \ldots\right), \ h = 1, \cdots, H.$

(3) Matrix in Transformer: Define the following matrix $C$ related to the attention layer

Softmax kernel:

$$C_{ih} = \frac{X_i W_h^Q \left(X_i W_h^K\right)^\top}{\sqrt{d}}, \ S_{ih} = \text{Softmax}(C_{ih}) \tag{17}$$

Gaussian kernel:

$$(C_{ih})_{kj} = -\frac{\left\| X_{ik \cdot} W_h^Q - X_{ij \cdot} W_h^K \right\|^2}{2\sqrt{d}}, \ (S_{ih})_{kj} = \exp\left((C_{ih})_{kj}\right) \tag{18}$$

$$C_i = [C_{i1}, \cdots, C_{iH}], \ S_i = [S_{i1}, \cdots, S_{iH}] \tag{19}$$

Define matrix $V_i'$ for each data $X_i$:

$$V_i' = \begin{bmatrix} X_i W_1^V & & \\ & \ddots & \\ & & X_i W_H^V \end{bmatrix}_{Hn \times d} , \ V = [V_1^\top, \cdots, V_N^\top]^\top. \tag{20}$$

Next, let us introduce several useful lemma which leads to Theorem 2:

## 1.2 Lemmas of Theorem 2

**Lemma 1.**

$$(1) \ \frac{\partial f(M; X)}{\partial W^V} = B^\top \left(\text{MH}(M; X) - y\right)\left(W^O\right)^\top \tag{21}$$

$$(2) \ \text{vec}\left(\frac{\partial f(M; X)}{\partial W^V}\right) = \left\langle \left(W^O\right)^\top \otimes B, \text{vec}(\text{MH}(M; X) - y) \right\rangle$$

$$= \left(\mathbb{I}_{Hd} \otimes B^\top\right) \cdot \left(W^O \otimes \mathbb{I}_N\right) \cdot (\text{MH}(M; X) - y) \tag{22}$$

$$(3) \ \frac{\partial f(M; X)}{\partial W_h^Q} = \frac{1}{\sqrt{d}} X^\top \mathbb{P}_h \frac{\partial f(M; X)}{\partial C} X W_h^K = \sum_{i=1}^N \frac{1}{\sqrt{d}} X_i^\top \mathbb{P}_h \frac{\partial f(M; X_i)}{\partial C_i} X_i W_h^K \tag{23}$$

$$(4) \ \frac{\partial f(M; X_i)}{\partial C_i} = \left((\text{MH}(M; X_i) - y_i)\left(W^O\right)^\top \left(V_i'\right)^\top\right) \odot S_i \tag{24}$$

$$- \left(\left(\left((\text{MH}(M; X_i) - y_i)\left(W^O\right)^\top \left(V_i'\right)^\top\right) \odot S_i \odot \Upsilon\left((\exp C_i)\mathbb{E}\right)\right) \mathbb{E}^\top\right) \odot \exp C_i \tag{25}$$

$$(5) \ \frac{\partial f(M; X)}{\partial C} = \text{diag}\left(\frac{\partial f(M; X_1)}{\partial C_1}, \cdots, \frac{\partial f(M; X_N)}{\partial C_N}\right) \tag{26}$$

**Remark:** The above lemma derives the closed form of the gradient of objective over $W^V, W^Q$. Notice that we can derive the derivative of $W^K$ in the same way as $W^Q$ due to symmetry, so we do not include the derivation here. Some of the lemmas here refers https://say-hello2y.github.io/2022-09-07/attention-gradient

**Lemma 2.** *Consider updating $W^Q, W^K, W^V$ at iteration $t$. Suppose $\sigma_{\max}(W^Q)$, $\sigma_{\max}(W^K)$, $\sigma_{\max}(W^V)$ are bounded during in the optimization phase, then we have the following conclusion:*

$$(1)\ \|d(S_i)\|_F \leq \phi_i \|d(W^Q)\|_F, \quad where\ \phi_i = \frac{n}{\sqrt{d}} \|X_i\|_F^2 \sqrt{\sum_{h=1}^{H} \sigma_{\max}^2\left(W_h^K\right)} \tag{27}$$

$$(2)\ \|d(S_i)\|_F \leq \psi_i \|d(W^K)\|_F, \quad where\ \psi_i = \frac{n}{\sqrt{d}} \|X_i\|_F^2 \sqrt{\sum_{h=1}^{H} \sigma_{\max}^2\left(W_h^Q\right)}. \tag{28}$$

$$(3)\ \|d(S_i)\|_F \leq \sqrt{\phi_i^2 + \psi_i^2} \cdot \|d(W^Q), d(W^K)\|_F. \tag{29}$$

$(4)\ \|\dfrac{\partial f(M; X_i)}{\partial W^Q}\|_F \leq Q_i \|\mathsf{MH}(M; X_i) - y_i\|_F,$

$$where\ Q_i = n\sqrt{H} \|X_i\|_F^3 \left\|W^O\right\|_2 \sqrt{\sum_{h=1}^{H} \sigma_{\max}^2\left(W_h^K\right)} \cdot \sigma_{\max}\left(W^V\right). \tag{30}$$

$(5)\ \|\dfrac{\partial f(M; X_i)}{\partial W^K}\|_F \leq K_i \|\mathsf{MH}(M; X_i) - y_i\|_F,$

$$where\ K_i = n\sqrt{H} \|X_i\|_F^3 \left\|W^O\right\|_2 \sqrt{\sum_{h=1}^{H} \sigma_{\max}^2\left(W_h^Q\right)} \cdot \sigma_{\max}\left(W^V\right). \tag{31}$$

**Lemma 3.** *Consider updating $W^Q, W^K, W^V$ at iteration $t$. Suppose $\sigma_{\max}(W^Q)$, $\sigma_{\max}(W^K)$, $\sigma_{\max}(W^V)$ are bounded during in the optimization phase, then we have the following conclusion:*

$(1)\ \|\mathsf{MH}(M_{t+1}; X) - \mathsf{MH}(M_t; X)\|_F \leq Z\|M_{t+1} - M_t\|_F,$ *where $Z$ is some positive constant.*

$$\tag{32}$$

$(2)\ \|\nabla f\left(M_{t+1}; X\right) - \nabla f\left(M_t; X\right)\|_2 \leq G\|M_{t+1} - M_t\|_F,$ *where $G$ is some positive constant.*

$$\tag{33}$$

**Lemma 4.** *Let $f : \mathbb{R}^n \to \mathbb{R}$ be a second order differentiable function. Let $x, y \in \mathbb{R}^n$ be given, and assume that $\|\nabla f(z) - \nabla f(x)\|_2 \leq C\|z - x\|_2$ for every $z = x + t(y - x)$ with $t \in [0, 1]$. Then,*

$$f(y) \leq f(x) + \langle \nabla f(x), y - x \rangle + \frac{C'}{2}\|x - y\|^2.$$

**Lemma 5.** *For matrix $A \in \mathbb{R}^{k \times l}, B \in \mathbb{R}^{l \times m}, C \in \mathbb{R}^{m \times n}$.*

$$\text{vec}(ABC) = (I_n \otimes AB)\,\text{vec}(C) = \left(C^{\mathsf{T}}B^{\mathsf{T}} \otimes I_k\right)\text{vec}(A)$$

$$\text{vec}(AB) = (I_m \otimes A)\,\text{vec}(B) = \left(B^{\mathsf{T}} \otimes I_k\right)\text{vec}(A)$$

$$\text{vec}(A \odot B) = \text{vec}(A) \odot \text{vec}(B).$$

## 1.3 Proof of Theorem 2

**Proof Sketch of Theorem 2:**
The main idea of the proof follows from [Nguyen and Mondelli, 2020]. Let us first recall a few notations. $\bar{\lambda}^V := \frac{2}{3}\left(1 + \sigma_{\max}(W_0^V)\right)$, $\underline{\lambda}^B := \sigma_{\min}(B_0)$. Using GD update rule, we aim to iteratively show

$$\begin{cases} \sigma_{\max}(W_r^V) \leq \frac{3}{2}\bar{\lambda}^V, r \in \{0, \ldots, t\}, \\ \sigma_{\max}(W_r^Q) \leq \frac{3}{2}\bar{\lambda}^Q, r \in \{0, \ldots, t\}, \\ \sigma_{\max}(W_r^K) \leq \frac{3}{2}\bar{\lambda}^K, r \in \{0, \ldots, t\}, \\ \sigma_{\min}\left(B_r\right) \geq \frac{1}{2}\underline{\lambda}^B, r \in \{0, \ldots, t\}, \\ f\left(M_r; X\right) \leq (1 - \eta\mu)^r f\left(M_0, X\right), r \in \{0, \ldots, t\} \end{cases} \tag{34}$$

Denote $\mu := \frac{1}{4}(\underline{\lambda}^B)^2\|W^O\|_2^2$. Let us discuss about the value of $\mu$. We know $W^O \in \mathbb{R}^{Hd\times 1}$, $B_0^\top \in \mathbb{R}^{HD\times Nn}$. We require $\mu > 0$, i.e, $\underline{\lambda}^B > 0$, which implies $B$ has full row rank. For simplicity, let us consider the $H = 1$ case. Recall the definition of $B$:

$$B := \begin{pmatrix} S_{11}X_1 \\ \cdots \\ S_{N1}X_N \end{pmatrix}$$

Suppose we initialize $W_1^Q, W_1^K$ such that each $S_{i1} \in \mathbb{R}^{n\times n}$ is full rank, then we can easily show that $\operatorname{rank}(S_{i1}X_i) = n$ if $X_i$ has full row rank. Suppose embedding dimension $D$ is large, with certain assumption on $X$, we can show $B$ has full row rank. For example, if each $X_i$ follows standard Gaussian distribution with $D >> N$, then $\operatorname{rank}(B) = Nn$ with probability 1 if we initialize $W_1^Q, W_1^K$ such that $S_{i1}$ is full rank.

Further, let us assume that $\sum_{h=1}^H (\bar{\lambda}_h^Q)^2 > 1$, $\sum_{h=1}^H (\bar{\lambda}_h^K)^2 > 1$, and initialization condition satisfies:

$$\frac{54n^2\sqrt{NH}\|X\|_F^6 \bar{\lambda}^V \left( \sum_{h=1}^H (\bar{\lambda}_h^Q)^2 + (\bar{\lambda}_h^K)^2 \right)}{(\underline{\lambda}^B)^2\|W^O\|_2 \min \left( \bar{\lambda}_h^Q, \bar{\lambda}_h^K, 1, \underline{\lambda}^B \right)} \leq 1 \tag{35}$$

**Remark 5.** *The initialization condition can be satisfied if $\|W^O\|_2$ is large and $\sigma_{\max}(W^V)$ is small. $\nu$ in Equation (8) is $\frac{1}{54}$.*

It is clear that Equation (34) holds when $t = 0$. Suppose it holds at iteration $t$, we prove it holds at iteration $t+1$.

$$\left\| W_{r+1}^V - W_0^V \right\|_F \overset{(i)}{\leq} \sum_{s=0}^r \left\| W_{r+1}^V - W_r^V \right\|_F = \eta \sum_{s=0}^r \left\| \nabla_{W^V} f\left(M_t; X\right) \right\|_F$$

$$\overset{(ii)}{\leq} \eta \sum_{s=0}^r \|B_r\|_F \|W^O\|_2 \left\| \mathsf{MH}(M_r; X) - y \right\|_2 \overset{(iii)}{\leq} \eta \|B_r\|_F \|W^O\|_2 \sum_{s=0}^r (1-\eta\mu)^{s/2} \left\| \mathsf{MH}(M_0; X) - y \right\|_2,$$

where (i) uses the triangle inequality; (ii) plugs in the expression of $\nabla_{W^V} f\left(M_t; X\right)$ and uses the Cauchy-Schwartz inequality; (iii) is because we assume the loss function $f(\cdot)$ linearly decreases until $t$-th iteration. Let $u = \sqrt{1-\eta\mu}$. So we have

$$\eta \|B_r\|_F \|W^O\|_2 \sum_{s=0}(1-\eta\mu)^{s/2} \left\| \mathsf{MH}\left(M_0; X\right) - y \right\|_2$$

$$\leq \frac{1}{\mu}\|B_r\|_F \|W^O\| \frac{1-u^{r+1}}{1-u}(1-u^2) \left\| \mathsf{MH}\left(M_0; X\right) - y \right\|_2$$

$$= \frac{1}{\mu}\| [S_{r,1}X_1, \cdots, S_{r,N}X_N] \|_F \|W^O\| \frac{1-u^{r+1}}{1-u}(1-u^2) \left\| \mathsf{MH}\left(M_0; X\right) - y \right\|_2 \tag{36}$$

$$\overset{(i)}{\leq} \frac{2n\sqrt{HN}}{\mu}\|X\|_F \|W^O\|_2 \left\| \mathsf{MH}\left(M_0; X\right) - y \right\|_F \overset{(ii)}{\leq} 1, \tag{37}$$

where (i) is because each element in $S_{r,i}$ has magnitude at most 1 and $\|S_{r,i}\|_F \leq n\sqrt{H}$, then by Cauchy-Schwartz inequality, we have $\|B\|_r \leq \sqrt{HN}\|X\|_F$; (ii) is due to the initialization condition. Then by Weyl's inequality, there is

$$\sigma_{\max}\left(W_{r+1}^V\right) \leq \sigma_{\max}(W_0^V) + 1 = \frac{3}{2}\bar{\lambda}^V.$$

Similarly, let us derive the upper bound for $\sigma_{\max}(W_{h,r}^Q)$.

$$\left\| W_{h,r+1}^Q - W_{h,0}^Q \right\|_F \overset{(i)}{\leq} \sum_{s=0}^{r} \left\| W_{h,r+1}^Q - W_{h,r}^Q \right\|_F = \eta \sum_{s=0}^{r} \left\| \nabla_{W_h^Q} f\left( M_t; X \right) \right\|_F$$

$$\overset{(ii)}{\leq} \eta \sum_{s=0}^{r} \sqrt{\sum_{i=1}^{N} Q_i^2} \left\| \mathsf{MH}\left( M_r; X \right) - y \right\|_2 \overset{(iii)}{\leq} \eta \sqrt{\sum_{i=1}^{N} Q_i^2} \sum_{s=0}^{r} (1 - \eta\mu)^{s/2} \left\| \mathsf{MH}\left( M_0; X \right) - y \right\|_2$$

$$\leq \frac{\sqrt{\sum_{i=1}^{N} Q_i^2}}{\mu} \frac{1 - u^{r+1}}{1 - u} \left( 1 - u^2 \right) \left\| \mathsf{MH}\left( M_0; X \right) - y \right\|_2$$

$$\leq \frac{2\sqrt{\sum_{i=1}^{N} Q_i^2}}{\mu} \left\| \mathsf{MH}\left( M_0; X \right) - y \right\|_2 \overset{(iv)}{\leq} \frac{1}{2} \bar{\lambda}_h^Q,$$

where (i) uses triangle inequality; (ii) uses Lemma 2 (4); (iii) comes from the assumption that loss function $f(\cdot)$ linearly decreases until $t$-th iteration; (iv) is due to the initialization condition Equation (35). Similarly, we can show

$$\eta \sqrt{\sum_{i=1}^{N} K_i^2} \sum_{s=0}^{r} (1 - \eta\mu)^{s/2} \left\| \mathsf{MH}\left( M_0; X \right) - y \right\|_2$$

$$\leq \frac{2\sqrt{\sum_{i=1}^{N} K_i^2}}{\mu} \left\| \mathsf{MH}\left( M_0; X \right) - y \right\|_2 \leq \frac{1}{2} \bar{\lambda}_h^K. \tag{38}$$

Then by Weyl's inequality, there is

$$\sigma_{\max}\left( W_{h,t+1}^Q \right) \leq \sigma_{\max}(W_{h,0}^Q) + \frac{1}{2} \bar{\lambda}_h^Q = \frac{3}{2} \bar{\lambda}_h^Q; \ \sigma_{\max}\left( W_{h,t+1}^K \right) \leq \sigma_{\max}(W_{h,0}^K) + \frac{1}{2} \bar{\lambda}_h^K = \frac{3}{2} \bar{\lambda}_h^K.$$

Now we aim to bound the eigenvalues of $B_{r+1}$.

$$\left\| B_{r+1} - B_0 \right\|_F \leq \sum_{s=0}^{r} \left\| B_{s+1} - B_s \right\|_F = \sum_{i=1}^{N} \sum_{s=0}^{r} \left\| S_{i,s+1} X_i - S_{i,s} X_i \right\|_F$$

$$\overset{(i)}{\leq} \sum_{i=1}^{N} \sum_{s=0}^{r} \| X_i \|_F \| S_{i,s+1} - S_{i,s} \|_F \overset{(ii)}{\leq} \sum_{i=1}^{N} \sum_{s=0}^{r} \| X_i \|_F \sum_{h=1}^{H} \| S_{ih,s+1} - S_{ih,s} \|_F$$

$$\overset{(iii)}{\leq} \eta \sum_{i=1}^{N} \sum_{s=0}^{r} \| X_i \|_F \cdot \sqrt{\phi_i^2 + \psi_i^2} \cdot \| \left( \nabla_{W^Q} f(M_s; X_i), \nabla_{W^K} f(M_s; X_i) \right) \|_F \quad,$$

$$\overset{(iv)}{\leq} \sum_{i=1}^{N} \sum_{s=0}^{r} \| X_i \|_F \cdot \sqrt{\phi_2^2 + \psi_2^2} \cdot \sqrt{Q_i^2 + K_i^2} \cdot \| \mathsf{MH}\left( M_s, X_i \right) - y_i \|_F$$

$$\overset{(v)}{\leq} \eta \sum_{s=0}^{r} \sqrt{\sum_{i=1}^{N} \| X_i \|_F^2 (\phi_i^2 + \psi_i^2)(Q_i^2 + K_i^2)(1 - \eta\mu)^{s/2} \| \mathsf{MH}\left( M_0; X \right) - y \|_2}$$

where (i) and (ii) uses triangle inequality and Cauchy-Schwartz inequality; (iii) comes from Lemma 2 (5); (iv) uses Lemma 2 and Cauchy-Schwartz inequality; (v) comes from Cauchy-Schwartz inequality. Together with our initialization condition, we have

$$\| B_{r+1} - B_0 \|_F \leq \frac{1}{\mu} \sqrt{\sum_{i=1}^{N} \| X_i \|_F^2 (\phi_i^2 + \psi_i^2)(Q_i^2 + K_i^2)} \cdot \frac{1 - u^{r+1}}{1 - u} \| \mathsf{MH}\left( M_0; X \right) - y \|_2$$

$$\leq \frac{2}{\mu} \sqrt{\sum_{i=1}^{N} \| X_i \|_F^2 (\phi_i^2 + \psi_i^2)(Q_i^2 + K_i^2)} \| \mathsf{MH}\left( M_0; X \right) - y \|_2 \overset{(i)}{\leq} \frac{1}{2} \underline{\lambda}^B,$$

where (i) comes from the initialization condition 35. By Weyl's inequality, we can derive the bound for the singular values of $B_t$:

$$\sigma_{\min}(B_{r+1}) \geq \sigma_{\min}(B_0) - \|B_{r+1} - B_0\|_F \geq \frac{1}{2}\lambda^B.$$

The final step is to show the last inequality holds. Since we have already showed $\sigma_{\max}(W_h^Q), \sigma_{\max}(W_h^K), \sigma_{\max}(W_h^V)$ are bounded, by Lemma 3 (2) we can conclude that:

$$\|\nabla f(M_{t+1}; X) - \nabla f(M_t)\|_2 \leq G\|M_{t+1} - M_t\|_F$$

Thus by Lemma 4, we choose $\eta < \frac{1}{2G}$, then the following hold true:

$$
\begin{aligned}
f(M_{t+1}; X) &= f(M_t - \eta \nabla f(M_t; X); X) \\
&\stackrel{(i)}{\leq} f(M_t; X) - \eta \|\nabla f(M_t; X)\|^2 + \frac{G}{2}\eta^2 \|\nabla f(M_t; X)\|^2 \\
&\stackrel{(ii)}{\leq} f(M_t; X) - \frac{1}{2}\eta \|\nabla f(M_t; X)\|^2 \\
&\stackrel{(iii)}{\leq} f(M_t; X) - \frac{1}{2}\eta \left\|\frac{\partial f(M_t; X)}{\partial W^V}\right\|^2 \\
&\stackrel{(iv)}{\leq} f(M_t; X) - \frac{1}{2}\eta \|W^O \otimes B_t^\top (\text{vec}(\text{MH}(M_t; X) - y))\|^2 \\
&\stackrel{(v)}{\leq} f(M_t; X) - \frac{1}{8}\eta \|W^O\|_2^2(\lambda^B)^2 \cdot f(M_t; X) \\
&= (1 - \frac{1}{4}\|W^O\|_2^2(\lambda^B)^2) \cdot f(M_t; X) \\
&\stackrel{(vi)}{=} (1 - \eta\mu)f(M_t; X),
\end{aligned}
$$

where (i) uses Lemma 4; (ii) is because we set $\eta < \frac{1}{2G}$; (iii) only considers the gradient over $W^V$; (iv) plugs in the closed form gradient in Lemma 1; (v) uses the property of smallest singular value and induction assumption; (vi) comes from the definition of $\mu$.

## 1.4 Lemma for Theorem 3

The following lemmas all consider the Transformers with Gaussian kernel attention 11.

**Lemma 6.**

$$(1) \quad \frac{\partial f(M; X)}{\partial W^V} = B^\top (\text{MH}(M; X) - y) (W^O)^\top \tag{39}$$

$$(2) \quad \text{vec}\left(\frac{\partial f(M; X)}{\partial W^V}\right) = \langle (W^O)^\top \otimes B, \text{vec}(\text{MH}(M; X) - y)\rangle$$

$$= (\mathbb{I}_{Hd} \otimes B^\top) \cdot (W^O \otimes \mathbb{I}_N) \cdot (\text{MH}(M; X) - y) \tag{40}$$

$$(3) \quad \frac{\partial f(M; X)}{\partial W_h^Q} = \frac{\partial f(M; X)}{\partial C} \cdot \frac{\partial C}{\partial W_h^Q} = \sum_{i=1}^N \frac{\partial f(M; X_i)}{\partial C_i} \cdot \frac{\partial C_i}{\partial W_h^Q} \tag{41}$$

$$(4) \quad \frac{\partial f(M; X_i)}{\partial C_i} = \left((\text{MH}(M; X_i) - y_i)(W^O)^\top (V_i')^\top\right) \odot S_i \tag{42}$$

$$(5) \quad \frac{\partial f(M; X)}{\partial C} = \left[\frac{\partial f(M; X_1)}{\partial C_1}, \cdots, \frac{\partial f(M; X_N)}{\partial C_N}\right]^\top \tag{43}$$

**Lemma 7.** *Consider updating $W^Q, W^K, W^V$ at iteration $t$. Suppose $\sigma_{\max}(W^Q)$, $\sigma_{\max}(W^K)$, $\sigma_{\max}(W^V)$ are bounded during in the optimization phase, then we have the following conclusion:*

$$(1) \|d(C_{ih})\|_F \leq \sqrt{\frac{2n}{d}} \|X_i\|_F^2 \sqrt{\sigma_{\max}^2\left(W_h^Q\right) + \sigma_{\max}^2\left(W_h^K\right)}\|d(W_h^Q)\|_F \tag{44}$$

$$(2) \left\|d\left(\frac{\partial C_{ih}}{\partial W_h^Q}\right)\right\|_F \leq \sqrt{n}\|X_i\|_F^2 \cdot \|d(W_h^Q)\|_F. \tag{45}$$

$$(3) \|\frac{\partial f(M; X_i)}{\partial C_i}\|_F \geq \min|V_i W^O| \cdot \min S_i \cdot \|\mathsf{MH}(M; X_i) - y_i\|_2, \tag{46}$$

$$\text{where } R_i = \left(\mathsf{MH}(M; X_i) - y_i\right)\left(W^O\right)^\top \left(V_i''\right)^\top. \tag{47}$$

$$(4) \left\|\frac{\partial f(M; X_i)}{\partial W_h^Q}\right\|_F \leq Q_i'\|\mathsf{MH}(M; X_i) - y_i\|_2, \tag{48}$$

$$Q_i' = \sqrt{\frac{2n}{d}} \|X_i\|_F^3 \|W^O\|_2 \sigma_{\max}(W^V)\sqrt{\sigma_{\max}^2\left(W_h^Q\right) + \sigma_{\max}^2\left(W_h^K\right)} \tag{49}$$

*where $\min|V'W^O|$ is the smallest absolute value of each element in vector $V'W^O$; $\min S$ is the smallest element in matrix $S$.*

**Lemma 8.** *Consider updating $W^Q, W^K, W^V$ at iteration $t$. Suppose $\sigma_{\max}(W^Q)$, $\sigma_{\max}(W^K)$, $\sigma_{\max}(W^V)$ are bounded during in the optimization phase, then we have the following conclusion:*

$$\|\mathsf{MH}(M_{t+1}; X) - \mathsf{MH}(M_t; X)\|_F \leq Z'\|M_{t+1} - M_t\|_F, \text{where } Z' \text{ is some positive constant.} \tag{50}$$

$$\|\nabla f(M_{t+1}; X) - \nabla f(M_t; X)\|_2 \leq G'\|M_{t+1} - M_t\|_F, \text{where } G' \text{ is some positive constant.} \tag{51}$$

## 1.5 Proof Sketch of Theorem 3.

(1)Using GD update rule, we aim to iteratively show

$$\begin{cases} \sigma_{\max}(W_r^Q) \leq \frac{3}{2}\bar{\lambda}^Q, r \in \{0, \ldots, t\}, \\ \sigma_{\min}\left(\frac{\partial C_h(M_r)}{\partial W_h^Q}\right) \geq \frac{1}{2}\delta, r \in \{0, \ldots, t\}, \\ \min S_r \geq \kappa, r \in \{0, \ldots, t\}, \\ f(M_r; X) \leq (1 - \eta\gamma)^r f(M_0, X), \ r \in \{0, \ldots, t\} \end{cases} \tag{52}$$

Denote $\gamma := \frac{1}{2}\delta^2\kappa^2\left(\min|V'W^O|\right)^2$. Let us discuss about the value of $\gamma$. We know $W^O \in \mathbb{R}^{Hd^V \times 1}$, $B_0^\top \in \mathbb{R}^{HD \times Nn}$, where $Hd > 1, HD > Nn$. We require $\gamma > 0$, i.e, $\delta > 0, \kappa > 0, \min|V'W^O| > 0$. It is clear that $\kappa > 0$ can hold as long as $W_h^Q$ is bounded. And it is easy to show that if $X_i \neq \mathbf{0}$, we can always choose $W^V$ and $W^O$, such that $\min|V'W^O| > 0$. Since $\frac{\partial C_h(M)}{\partial W_h^Q} \in \mathbb{R}^{Nn^2 \times Dd}$, suppose we initialize $W_h^Q, W_h^K$ such that $\text{rank}(\frac{\partial C_h(M_0)}{\partial W_h^Q}) = Nn^2$, then we have $\sigma_{\min}\left(\frac{\partial C_h(M_0)}{\partial W_h^Q}\right) \geq \delta$ for some positive constant $\delta$. Further, we assume the initialization condition satisfies:

$$\frac{8n\|X\|_F^5\|W^O\|_2\bar{\lambda}^V(\bar{\lambda}_h^Q + \bar{\lambda}_h^K)\exp\left(\frac{9}{4}\|X\|_F^2\left(\left(\bar{\lambda}_h^Q\right)^2 + (\bar{\lambda}_h^K)^2\right)\right)}{\delta^2\left(\min(|V'W^O|)\right)^2 \cdot \min\left(\delta, \bar{\lambda}_h^Q\right)}\|\mathsf{MH}(M_0; X) - y\|_2 \leq 1 \tag{53}$$

**Remark 6.** *The initialization condition can be satisfied if $\|W^O\|_2$ is large and $\sigma_{\max}(W^V)$ is small. $\nu'$ in Equation (12) is $\frac{1}{8}$.*

Similar to the proof of Theorem 2, we use induction to prove the theorem. Equation (52) holds when $t = 0$. Suppose it holds at iteration $t$, we prove it holds at iteration $t + 1$.

$$\left\|W_{h,r+1}^Q - W_{h,0}^Q\right\|_F \overset{(i)}{\leq} \sum_{s=0}^r \left\|W_{h,r+1}^Q - W_{h,r}^Q\right\|_F = \eta \sum_{s=0}^r \left\|\nabla_{W_h^Q} f\left(M_t; X\right)\right\|_F$$

$$\overset{(ii)}{\leq} \eta \sum_{s=0}^r \sqrt{\sum_{i=1}^N Q_i'^2} \left\|\mathsf{MH}(M_r; X) - y\right\|_2 \overset{(iii)}{\leq} \eta \sqrt{\sum_{i=1}^N Q_i'^2} \sum_{s=0}^r (1 - \eta\gamma)^{s/2} \left\|\mathsf{MH}(M_0; X) - y\right\|_2,$$

where (i) uses triangle inequality; (ii) comes from Lemma 7 and Cauchy-Schwartz inequality; (iii) is from the induction assumption that loss function $f(\cdot)$ linearly decreases until $t$-th iteration. Let $u = \sqrt{1 - \eta\gamma}$. So we have

$$\left\|W_{h,r+1}^Q - W_{h,0}^Q\right\|_F \leq \eta \sqrt{\sum_{i=1}^N Q_i'^2} \sum_{s=0}^r (1 - \eta\gamma)^{s/2} \left\|\mathsf{MH}\left(M_0; X\right) - y\right\|_2 \tag{54}$$

$$\leq \frac{1}{\gamma} \sqrt{\sum_{i=1}^N Q_i'^2} \frac{1 - u^{r+1}}{1 - u} (1 - u^2) \left\|\mathsf{MH}(M_0; X) - y\right\|_2$$

$$\leq \frac{2\sqrt{\sum_{i=1}^N Q_i'^2}}{\gamma} \left\|\mathsf{MH}(M_0; X) - y\right\|_F \overset{(i)}{\leq} \frac{1}{2}\bar{\lambda}_h^Q, \tag{55}$$

where (i) comes from the initialization condition. Then by Weyl's inequality, there is

$$\sigma_{\max}\left(W_{h,t+1}^Q\right) \leq \sigma_{\max}(W_{h,0}^Q) + \frac{1}{2}\bar{\lambda}_h^Q = \frac{3}{2}\bar{\lambda}_h^Q.$$

$$\left\|\frac{\partial C_h\left(M_{r+1}\right)}{\partial W_h^Q} - \frac{\partial C_h\left(M_0\right)}{\partial W_h^Q}\right\|_F \overset{(i)}{\leq} \sum_{s=0}^r \left\|\frac{\partial C_h\left(M_{s+1}\right)}{\partial W_h^Q} - \frac{\partial C_h\left(M_s\right)}{\partial W_h^Q}\right\|_F$$

$$\overset{(ii)}{\leq} \eta\sqrt{n}\|X\|_F^2 \sum_{s=0}^r \left\|\nabla_{W_h^Q} f\left(M_s; X\right)\right\|_F$$

$$\overset{(iii)}{\leq} \eta\sqrt{n}\|X\|_F^2 \sum_{s=0}^r \sqrt{\sum_{i=1}^N Q_i'^2} \left\|\mathsf{MH}\left(M_s; X\right) - y\right\|_2$$

$$\overset{(iv)}{\leq} \eta\sqrt{n}\|X\|_F^2 \sqrt{\sum_{i=1}^N Q_i'^2} \sum_{s=0}^r (1 - \eta\gamma)^{s/2} \left\|\mathsf{MH}\left(M_0; X\right) - y\right\|_2,$$

$$\leq \frac{2}{\gamma}\sqrt{n}\|X\|_F^2 \sqrt{\sum_{i=1}^N Q_i'^2} \left\|\mathsf{MH}\left(M_0; X\right) - y\right\|_2$$

$$\overset{(v)}{\leq} \frac{1}{2}\delta,$$

where (i) uses triangle inequality; (ii) applies Lemma 7 (2) and Cauchy-Schwartz inequality; (iii) uses Lemma 7 (4); (iv) applies the induction assumption that the loss function $f(\cdot)$ linearly decreases until $t$-th iteration; (v) comes from the initialization condition. Then by Weyl's inequality, there is

$$\sigma_{\max}\left(\frac{\partial C_h\left(M_{t+1}\right)}{\partial W_h^Q}\right) \geq \sigma_{\max}\left(\frac{\partial C_h\left(M_0\right)}{\partial W_h^Q}\right) - \frac{1}{2}\delta = \frac{1}{2}\delta.$$

For each element in $S_{ih}$, we have close form

$$S\left(W_h^Q, W_h^K; X_i\right)_{kj} = \exp\left(-\frac{1}{2\sqrt{d}}\left\|X_{ik} \cdot W_h^Q - X_{ij} \cdot W_h^K\right\|^2\right)$$

Since we have already showed that $\sigma_{\max}\left(W_{h,r}^Q\right) \leq \frac{3}{2}\bar{\lambda}_h^Q$, it follows directly each element in matrix $S_t$ is lower bounded by some constant $\kappa$ for any $t$. Now we derive the expression of $\kappa$:

$$\exp\left(-\frac{1}{2\sqrt{d}}\left\|X_{ik}\cdot W_{h,t}^Q - X_{ij}\cdot W_h^K\right\|^2\right)$$
$$\overset{(i)}{\geq} \exp\left(-\frac{1}{\sqrt{d}}\left(\|X_{ik}\cdot W_{h,t}^Q\|^2 + \|X_{ij}\cdot W_h^K\|^2\right)\right)$$
$$\overset{(ii)}{\geq} \exp\left(-\frac{1}{\sqrt{d}}\left(\frac{9}{4}(\bar{\lambda}_h^Q)^2\|X_{ik\cdot}\|^2 + (\bar{\lambda}_h^K)^2\|X_{ij\cdot}\|^2\right)\right)$$
$$\overset{(iii)}{\geq} \exp\left(-\frac{9}{4}\|X\|_F^2\left((\bar{\lambda}_h^Q)^2 + (\bar{\lambda}_h^K)^2\right)\right)$$
$$:= \kappa,$$

where (i) uses Cauchy-Schwartz inequality; (ii) applies the induction assumption $\sigma_{\max}(W_{h,t}^Q) \leq \frac{3}{2}\bar{\lambda}_h^Q$ and property of singular value; (iii) is because $d \geq 1$. Thus, we have $\min S_t \geq \kappa$. Finally, we aim to show $f\left(M_{t+1};X\right) \leq (1-\eta\gamma)f\left(M_t,X\right)$. By Lemma 8, since we have showed that $\sigma_{\max}(W_h^Q)$ is bounded, we can directly derive that

$$\|\nabla f\left(M_{t+1};X\right) - \nabla f\left(M_t;X\right)\|_2$$
$$= \left\|\nabla_{W_h^Q} f\left(M_{t+1};X\right) - \nabla_{W_h^Q} f\left(M_t;X\right)\right\|_2$$
$$\leq G'\|M_{t+1} - M_t\|_F$$

Finally, by Lemma 4, choose $\eta < \frac{1}{2G'}$, we have the following holds:

$$f\left(M_{t+1};X\right) = f\left(M_t - \eta\nabla f(M_t;X);X\right)$$
$$\overset{(i)}{\leq} f\left(M_t;X\right) - \eta\|\nabla_{W^Q} f\left(M_t;X\right)\|^2 + \frac{G'}{2}\eta^2\|\nabla_{W^Q} f\left(M_t;X\right)\|^2$$
$$\overset{(ii)}{\leq} f\left(M_t;X\right) - \frac{1}{2}\eta\left\|\nabla_{W_h^Q} f\left(M_t;X\right)\right\|^2$$
$$\overset{(iii)}{=} f\left(M_t;X\right) - \frac{1}{2}\eta\left\|\frac{\partial f(M_t;X)}{\partial C(M_t)}\cdot\left(\frac{\partial C(M_t)}{\partial W_h^Q}\right)\right\|_F^2$$
$$\overset{(iv)}{\leq} f\left(M_t;X\right) - \frac{1}{4}\eta\delta^2\left\|\frac{\partial f\left(M_t;X\right)}{\partial C\left(M_t\right)}\right\|_F^2$$
$$\overset{(v)}{\leq} f\left(M_t;X\right) - \frac{1}{4}\eta\delta^2\left\|\left((\mathsf{MH}\left(M;X\right) - y)\left(W^O\right)^\top (V')^\top\right)\odot S\right\|_F^2$$
$$\overset{(vi)}{\leq} f\left(M_t;X\right) - \frac{1}{4}\eta\delta^2\kappa^2\cdot(\min|V'W^O|)^2\|\mathsf{MH}\left(M_0;X\right) - y\|_2^2$$
$$\overset{(vii)}{=} (1-\eta\gamma)f(M_t;X),$$

where (i) uses Lemma 4 (2); (ii) is because we choose $\eta < \frac{1}{2G'}$; (iii) writes down the expression of gradient according to chain rule in Lemma 6; (iv) uses the induction assumption $\sigma_{\max}\left(\frac{\partial C_h(M_{t+1})}{\partial W_h^Q}\right) \geq \frac{1}{2}\delta$ and property of singular value; (v) uses Lemma 6 (4); (vi) comes from Lemma 7 (3); (vii) uses the definition of $\gamma$.

(2)Next, we show the convergence result for Transformer with Softmax kernel with only $W^Q$ updated. Since we assume parameters are all bounded during optimization phase, by Lemma 8, we can easily show that there exists constant $G'$ (see xx for details), such that

$$\left\|\nabla_{W_h^Q} f\left(M_{t+1};X\right) - \nabla_{W_h^Q} f\left(M_t;X\right)\right\|_2 \leq G'\|M_{t+1} - M_t\|_F \tag{56}$$

Then by Lemma 4, choose $\eta' < \frac{1}{2G'}$ we have

$$f\left(M_{t+1}; X\right) = f\left(M_t - \eta \nabla f\left(M_t; X\right); X\right)$$

$$\leq f\left(M_t; X\right) - \eta' \left\|\nabla f\left(M_t; X\right)\right\|^2 + \frac{G'}{2}\eta'^2 \left\|\nabla f\left(M_t; X\right)\right\|^2$$

$$\leq f\left(M_t; X\right) - \frac{1}{2}\eta' \left\|\nabla f\left(M_t; X\right)\right\|^2$$

## 1.6 Proof of Lemma in Section 1.2

**Proof of Lemma 2 (1).**

*Proof.* **Step 1:** When $W^Q, W^K$ are updated, we aim to prove

$$\|d(S_i)\|_F \leq n\|d(C_i)\|_F.$$

**Step 2:** We aim to show $\|d(C_i)\|_F \leq \frac{n}{\sqrt{d}} \|X_i\|_F^2 \sqrt{\sum_{h=1}^{H} \sigma_{\max}^2\left(W_h^K\right)} \cdot \left\|d\left(W^Q\right)\right\|_F$. Combine the
above two steps, we can derive the bound in Equation (27).
Proof of Step 1: First, we can write down the closed form of the differential of $S_i$:

$$\|d(S_i)\|_F = \|S_i \odot d(C_i) - S_i \odot \Upsilon((\exp C_i)\mathbb{E}) \odot d(\exp(C_i)\mathbb{E}))\|_F \tag{57}$$

We reorganize the terms on the right side of Equation (57), we have the following equation:

$$\|d(S_i)\|_F = \|S_i \odot \left(d(C_i) - \Upsilon((\exp C_i)\mathbb{E}) \odot d((\exp C_i)\mathbb{E})\right)\|_F$$

$$= \|S_i \odot \left(d(C_i) - \Upsilon((\exp C_i)\mathbb{E}) \odot ((\exp C_i) \odot d(C_i))\,\mathbb{E}\right)\|_F \tag{58}$$

Since $C_i = [C_{i1}, \cdots, C_{iH}]$, we will investigate each $C_{ih}$, $h = 1, 2, \cdots, H$. We focus on the term
$d(C_i) - \Upsilon((\exp C_i)\mathbb{E}) \odot (\exp(C_i) \odot d(C_i))\,\mathbb{E}$ in Equation (58). We write down the close form of
the element in the $k$-th row and $j$-th column:

$$[d(C_{ih}) - \Upsilon(\exp(C_{ih})\mathbb{E}) \odot (\exp(C_{ih}) \odot d(C_{ih}))\,\mathbb{E}]_{kj} \tag{59}$$

$$\overset{(i)}{=} \left(1 - \frac{\exp\left(C_{ihkj}\right)}{\sum_{j=1}^{n} \exp\left(C_{ihkj}\right)}\right) d(C_{ihkj}) - \frac{\sum_{p \neq j} \exp\left(C_{ihkp}\right) d(C_{ihkp})}{\sum_{j=1}^{n} \exp\left(C_{ihkj}\right)} \tag{60}$$

$$\overset{(ii)}{\leq} \sqrt{\left(1 - \frac{\exp\left(C_{ihkj}\right)}{\sum_{j=1}^{n} \exp\left(C_{ihkj}\right)}\right)^2 + \sum_{p \neq j}\left(\frac{\exp(C_{ihkp})}{\sum_{j=1}^{n} \exp\left(C_{ihkj}\right)}\right)^2} \cdot \sqrt{\sum_{j=1}^{n}\left(d(C_{ihkj})\right)^2} \tag{61}$$

$$\overset{(iii)}{\leq} \sqrt{n}\|d(C_{ihk})\|_F, \tag{62}$$

where (i) is expand the closed form of Equation (59); (ii) uses the Cauchy-Schwartz inequality; (iii)
is because each element in the square root in (ii) is upper bounded by 1. With Equation (61), we can
easily show

$$\|d(C_{ih}) - \Upsilon((\exp C_{ih})\mathbb{E}) \odot ((\exp C_{ih}) \odot d(C_{ih}))\,\mathbb{E}\|_F \leq \sqrt{n}\sqrt{\sum_{k=1}^{n}\sum_{j=1}^{n}\|d(C_{ihk})\|_F^2} \leq n\|d(C_{ih})\|_F \tag{63}$$

Since every element in $S_i$ has magnitude less than 1, we have

$$\|d(S_i)\|_F = \|S_i \odot \left(d\left(C_i\right) - \Upsilon\left(\left(\exp C_i\right)\mathbb{E}\right) \odot \left(\left(\exp C_i\right) \odot d\left(C_i\right)\right)\mathbb{E}\right)\|_F \tag{64}$$

$$\leq \|d(C_{ih}) - \Upsilon((\exp C_{ih})\mathbb{E}) \odot ((\exp C_{ih}) \odot d(C_{ih}))\,\mathbb{E}\|_F \tag{65}$$

$$\overset{(i)}{\leq} n\sqrt{H}\|d(C_i)\|_F, \tag{66}$$

where (i) is from Cauchy-Schawatz inequality.

Proof of Step 2: We aim to show $\|d(C_i)\|_F \le \frac{n}{\sqrt{d}}\|X_i\|_F^2 \sqrt{\sum_{h=1}^{H} \sigma_{\max}^2 (W_h^K)} \cdot \|d(W^Q)\|_F$. Similarly, we investigate $\|d(C_{ih})\|_F, \; h = 1, 2, \cdots, H$. We have

$$\|d(C_{ih})\|_F = \left\| \frac{X_i d(W_h^Q)(X_i W_h^K)^\top}{\sqrt{d}} \right\|_F \le \frac{1}{\sqrt{d}}\|X_i\|_F^2 \sigma_{\max}(W_h^K)\|d(W_h^Q)\|_F \tag{67}$$

Then plug the above inequality to Equation (63), we can derive

$$\|d(S_{ih})\|_F \le \frac{n}{\sqrt{d}}\|X_i\|_F^2 \sigma_{\max}(W_h^K) \left\| d(W_h^Q) \right\|_F \tag{68}$$

Thus by Cauchy-Schwartz inequality, it is easy to show

$$\|d(S_i)\|_F \le \frac{n}{\sqrt{d}}\|X_i\|_F^2 \sqrt{\sum_{h=1}^{H} \sigma_{\max}^2 (W_h^K)} \cdot \|d(W^Q)\|_F .$$

$\square$

**Proof of Lemma 2 (4).**

*Proof.* We first write down the close form of gradient of $f(\cdot)$ over $W_h^Q$ by Lemma 1, and derive the upper bound of the norm of the gradient.

$$\left\| \frac{\partial f(M; X_i)}{\partial W_h^Q} \right\|_F = \left\| \frac{1}{\sqrt{d}} X_i^\top \frac{\partial f(M; X_i)}{\partial C_i} \mathbb{P}_h^\top X_i W_h^K \right\|_F \le \|X_i\|_F^2 \sigma_{\max}(W_h^K) \left\| \frac{\partial f(M; X_i)}{\partial C_i} \right\|_F \tag{69}$$

By Lemma 1, there is

$$\frac{\partial f(M; X_i)}{\partial C_i} = \left( (\mathsf{MH}(M; X_i) - y_i)(W^O)^\top (V_i')^\top \right) \odot S_i$$
$$- \left( \left( \left( (\mathsf{MH}(M; X_i) - y_i)(W^O)^\top (V_i')^\top \right) \odot S_i \odot \Upsilon\big((\exp C_i)\mathbb{E}\big) \right) \mathbb{E}^\top \right) \odot \exp C_i \tag{70}$$

Denote $R_i = (\mathsf{MH}(M; X_i) - y_i)(W^O)^\top (V_i')^\top, \; R_i = [R_{i1}, \cdots, R_{iH}]$. Write down the close form of the element in the $k$-th row and $j$-th column:

$$\left[ R_{ih} S_{ih} - \left( (R_{ih} \odot C_{ih} \odot \Upsilon\left((\exp C_{ih})\mathbb{E}\right)) \mathbb{E}^\top \right) \odot (\exp C_{ih}) \right]_{kj}$$

$$= R_{ihkj} S_{ihkj} - \frac{\exp(C_{ihkj}) \sum_{j=1}^{n} R_{ihkj} S_{ihkj}}{\sum_{j=1}^{n} \exp(C_{ihkj})}$$

$$= \left( S_{ihkj} - \frac{(\exp C_{ihkj}) S_{ihkj}}{\sum_{j=1}^{n} \exp(C_{ihkj})} \right) \cdot R_{ihkj} - \sum_{p \ne j} \frac{(\exp C_{ihkp}) S_{ihkj}}{\sum_{j=1}^{n} \exp(C_{ihkp})} R_{ihkp}$$

$$\overset{(i)}{\le} \sqrt{\left( 1 - \frac{\exp(C_{ihkj})}{\sum_{j=1}^{n} \exp(C_{ihkj})} \right)^2 + \sum_{p \ne j} \left( \frac{\exp(C_{ihkp})}{\sum_{j=1}^{n} \exp(C_{ihkj})} \right)^2} \cdot \|R_{ihk}\|_F$$

$$\overset{(ii)}{\le} \sqrt{n}\|R_{ihk}\|_F$$

where (1) is due to the Cauchy-Schwartz inequality; (ii) is because each element within the squre root term in (i) has magnitude at most 1. Thus, we can further derive

$$\left\|\frac{\partial f\left(M;X_i\right)}{\partial C_{ih}}\right\|_F = \left\|R_{ih}\odot S_{ih} - \left(\left(R_{ih}\odot S_{ih}\odot \Upsilon((\exp C_{ih})\mathbb{E})\right)\mathbb{E}^\top\right)\odot \exp C_{ih}\right\|_F$$

$$\overset{(i)}{\leq} \sqrt{n}\sum_{k=1}^{n}\sum_{j=1}^{n}\|R_{ihk}\|_F \overset{(ii)}{\leq} n\|R_{ih}\|_F$$

$$\overset{(iii)}{\leq} n\|X_i\|_F\|W^O\|_2\sigma_{\max}(W_h^V)\|\mathsf{MH}(M;X_i) - y_i\|_2,$$

where (i) if from the bound in Equation (71); (ii) comes from Cauchy-Schwatz inwquality; (iii) uses the property of Frobenious norm. Thus, by Cauchy-Schwartz inequality, we can derive the upper bound for $\left\|\frac{\partial f(M;X_i)}{\partial C_i}\right\|_F$:

$$\left\|\frac{\partial f\left(M;X_i\right)}{\partial C_i}\right\|_F \leq n\sqrt{H}\|X_i\|_F\|W^O\|_2\sigma_{\max}(W^V)\|\mathsf{MH}(M;X_i) - y_i\|_2 \tag{71}$$

So plug the above inequality into Equation (69), we can derive the upper bound for $\left\|\frac{\partial f(M;X_i)}{\partial W_h^Q}\right\|_F$:

$$\left\|\frac{\partial f\left(M;X_i\right)}{\partial W_h^Q}\right\|_F \leq \|X_i\|_F^2\,\sigma_{\max}\left(W_h^K\right)\left\|\frac{\partial f\left(M;X\right)}{\partial C_i}\right\|_F$$

$$\leq n\sqrt{H}\,\|X_i\|_F^3\,\|W^O\|_2\,\sigma_{\max}\left(W_h^K\right)\sigma_{\max}\left(W_h^V\right)\|\mathsf{MH}\left(M;X_i\right) - y_i\|_2$$

$$\leq n\sqrt{H}\,\|X_i\|_F^3\,\|W^O\|_2\,\sqrt{\sum_{h=1}^{H}\sigma_{\max}^2\left(W_h^K\right)}\sigma_{\max}\left(W^V\right)\|\mathsf{MH}\left(M;X_i\right) - y_i\|_2$$

$$\square$$

**Proof of Lemma 3 (1).** By Mean Value Theorem and Cauchy-Schwartz inequality,

$$|f\left(M_{t+1};X_i\right) - f\left(M_t;X_i\right)|$$

$$= \left\langle\frac{\partial f(M_t';X_i)}{\partial W}, M_{t+1} - M_t\right\rangle$$

$$\leq \sqrt{\left\|\frac{\partial f(M_t';X_i)}{\partial W^Q}\right\|^2 + \left\|\frac{\partial f(M_t';X_i)}{\partial W^K}\right\|^2 + \left\|\frac{\partial f(M_t;X_i)}{\partial W^V}\right\|^2}\|M_{t+1} - M_t\|_F, \tag{72}$$

where $M_t'$ is between $M_t$ and $M_{t+1}$. We can derive the upper bound of the norm of $\nabla_{W^V}f(M;X_i)$:

$$\left\|\frac{\partial f(M_t;X_i)}{\partial W^V}\right\|_F = \|B_i^\top\left(\mathsf{MH}(M_t;X_i) - y_i\right)\left(W^O\right)^\top\|_F$$

$$\leq \|B_i\|_F\|\mathsf{MH}(M_t;X_i) - y_i\|_F\|W^O\|_2$$

$$\leq n\sqrt{H}\|X_i\|_F\|W^O\|_2\|\mathsf{MH}(M_t;X_i) - y_i\|_F \tag{73}$$

By Lemma 2, we know

$$\left\|\frac{\partial f(M_t;X_i)}{\partial W^Q}\right\|_F \leq Q_i\,\|\mathsf{MH}\left(M;X_i\right) - y_i\|_2;\quad \left\|\frac{\partial f(M_t;X_i)}{\partial W^K}\right\|_F \leq K_i\,\|\mathsf{MH}\left(M;X_i\right) - y_i\|_2.$$

$$\|f\left(M_{t+1};X_i\right) - f\left(M_t;X_i\right)\|_2 \leq \sqrt{Q_i^2 + K_i^2 + n^2 H\sigma_{\max}^2(X_i)\|W^O\|^2}\|M_{t+1} - M_t\|_F$$

$$:= Z_i\|M_{t+1} - M_t\|_F \tag{74}$$

Therefore, together with Equation (73), we have

$$\|f\left(M_{t+1};X\right) - f\left(M_t;X\right)\|_2 \leq N\sqrt{\max_i Q_i^2 + \max_i K_i^2 + n^2 H\max_i\|X_i\|_F^2}\|M_{t+1} - M_t\|_F$$

$$:= Z\|M_{t+1} - M_t\|_F \tag{75}$$

**Proof of Lemma 3 (2).**

*Proof.* By triangle inequality, we have

$$\|\nabla_W f(M_{t+1}; X) - \nabla_W f(M_t; X)\|_F$$

$$\leq \|\nabla_{W^Q} f(M_{t+1}; X) - \nabla_{W^Q} f(M_{t+1}; X)\|_F + \|\nabla_{W^K} f(M_{t+1}; X) - \nabla_{W^K} f(M_{t+1}; X)\|_F$$

$$+ \|\nabla_{W^V} f(M_{t+1}; X) - \nabla_{W^V} f(M_{t+1}; X)\|_F \tag{76}$$

$$\leq \sum_{i=1}^{N} \big( \|\nabla_{W^Q} f(M_{t+1}; X) - \nabla_{W^Q} f(M_{t+1}; X)\|_F + \|\nabla_{W^K} f(M_{t+1}; X) - \nabla_{W^K} f(M_{t+1}; X)\|_F$$

$$\tag{77}$$

$$+ \|\nabla_{W^v} f(M_{t+1}; X) - \nabla_{W^v} f(M_{t+1}; X)\|_F \big) \tag{78}$$

**Step 1:** Derive upper bound for

$\|\nabla_{W^Q} f(M_{t+1}; X_i)) - \nabla_{W^Q} f(M_t; X_i))\|_F = \| \operatorname{vec}(\nabla_{W^Q} f(M_{t+1}; X_i)) - \operatorname{vec}(\nabla_{W^Q} f(M_t; X_i))\|_2.$
First, we give the vectorized expression of $\nabla_{W^Q} f(M_t; X_i)$. Recall we denote $U_i = \left( (\mathsf{MH}(M; X_i) - y_i) (W^O)^\top (V_i')^\top \right) \odot S_i$. By Lemma 1, we can derive the close form of $\operatorname{vec}(\nabla_{W^Q} f(M_t; X_i))$:

$$\operatorname{vec}(\nabla_{W^Q} f(M; X_i)) \overset{(i)}{=} \operatorname{vec}(U_i) - \operatorname{vec}\left( (U_i \odot \Upsilon((\exp C_i)\mathbb{E})) \mathbb{E}^\top \right) \odot \operatorname{vec}(\exp C_i)$$

$$\overset{(ii)}{=} \operatorname{vec}(U_i) - (\mathbb{E} \otimes \mathbb{I}_n) \operatorname{vec}\left( U_i \odot \Upsilon((\exp C_i)\mathbb{E}) \right) \odot \operatorname{vec}(\exp C_i)$$

$$\overset{(iii)}{=} \operatorname{vec}(U_i) - (\mathbb{E} \otimes \mathbb{I}_n) \operatorname{vec}(U_i) \odot \operatorname{vec}\left( \Upsilon((\exp C_i)\mathbb{E}) \right) \odot \operatorname{vec}(\exp C_i)$$

$$\overset{(iv)}{=} \operatorname{vec}(U_i) - (\mathbb{E} \otimes \mathbb{I}_n) \operatorname{vec}(U_i) \odot \operatorname{vec}(S_i)$$

$$\overset{(v)}{=} \mathbb{I}_{n^2 H} \operatorname{vec}(U_i) \odot \operatorname{vec}(\mathbf{1}_n \mathbf{1}_{nH}^\top) - (\mathbb{E} \otimes \mathbb{I}_n) \operatorname{vec}(U_i) \odot \operatorname{vec}(S_i)$$

$$\overset{(vi)}{=} \left( \mathbb{I}_{n^2 H} - (\mathbb{E} \otimes \mathbb{I}_n) \right) \operatorname{vec}(U_i) \odot \operatorname{vec}(\mathbf{1}_n \mathbf{1}_{nH}^\top - S_i), \tag{79}$$

where (i) uses the Lemma 1; (ii) and (iii) comes from the property of vectorization in Lemma 5; (vi) uses the definition of $S_i$ ; (v) gives an equivalent expression of $\operatorname{vec}(U_i)$; (vi) reorganizies (v). Further, it is easy to verify that:

$$\|U_i\|_F = \left\| \left( (\mathsf{MH}(M; X_i) - y_i) (W^O)^\top (V_i')^\top \right) \odot S_i \right\|_F \leq \|R_i\|_F$$

$$= (\|\mathsf{MH}(M; X_i)\|_2 + \|y_i\|_2) \|W^O\|_2 \|X_i\|_F \sigma_{\max}(W^V)$$

$$\leq \left( n\sqrt{H} \sigma_{\max}(W^V) \|X_i\|_F \|W^O\|_2 + \|y_i\|_2 \right) \|W^O\|_2 \|X_i\|_F \sigma_{\max}(W^V)$$

$$\leq \left( n\sqrt{H} \sigma_{\max}(W^V) \|X\|_F \|W^O\|_2 + \|y\|_2 \right) \|W^O\|_2 \|X\|_F \sigma_{\max}(W^V)$$

$$:= \bar{R} \tag{80}$$

Next, let us derive upper bound for $\|\nabla_{W^Q} f(M_{t+1}; X_i) - \nabla_{W^Q} f(M_{t+1}; X_i)\|_F$.

$$\|\nabla_{W^Q} f(M_{t+1}; X_i) - \nabla_{W^Q} f(M_{t+1}; X_i)\|_F$$

$$\overset{(i)}{=} \left\| \left( \mathbb{I}_{n^2 H} - (\mathbb{E} \otimes \mathbb{I}_n) \right) \left( \operatorname{vec}(U_{i,t+1}) \odot \operatorname{vec}(S_{i,t+1}) - \operatorname{vec}(U_{i,t}) \odot \operatorname{vec}(S_{i,t}) \right) \right\|_F$$

$$= \left\| \left( \mathbb{I}_{n^2 H} - (\mathbb{E} \otimes \mathbb{I}_n) \right) \left( \operatorname{vec}(U_{i,t+1}) \odot \operatorname{vec}(S_{i,t+1}) - \operatorname{vec}(U_t) \odot \operatorname{vec}(S_{i,t+1}) + \operatorname{vec}(U_{i,t}) \odot \operatorname{vec}(S_{i,t+1}) - \operatorname{vec}(U_{i,t}) \odot \operatorname{vec}(S \right. \right.$$

$$\overset{(ii)}{\leq} \|\mathbb{I}_{n^2 H} - (\mathbb{E} \otimes \mathbb{I}_n)\|_F \left( \| \operatorname{vec}(U_{i,t+1} - U_{i,t})\|_F + \|U_{i,t}\|_F \|S_{i,t+1} - S_{i,t}\|_F \right)$$

$$\overset{(iii)}{\leq} n\sqrt{H} \left( \|\operatorname{vec}(U_{i,t+1} - U_{i,t})\|_F + \bar{R} \|S_{i,t+1} - S_{i,t}\|_F \right)$$

$$\overset{(iv)}{=} n\sqrt{H} \left( \|R_{i,t+1} \odot S_{i,t+1} - R_{i,t} \odot S_{i,t}\|_F + \bar{R} \|S_{i,t+1} - S_{i,t}\|_F \right)$$

$$= n\sqrt{H} \left( \|(R_{i,t+1} \odot S_{i,t+1} - R_{i,t} \odot S_{i,t+1} + R_{i,t} \odot S_{i,t+1} - R_{i,t} \odot S_{i,t})\|_F + \bar{R} \|S_{i,t+1} - S_{i,t}\|_F \right)$$

$$\overset{(v)}{\leq} n\sqrt{H} \left( \|(R_{i,t+1} - R_{i,t}) \odot S_{i,t+1}\|_F + \|R_{i,t} \odot S_{i,t+1} - R_{i,t} \odot S_t)\|_F + \bar{R} \|S_{t+1} - S_t\|_F \right)$$

$$\overset{(vi)}{\leq} n\sqrt{H} \left( \|R_{i,t+1} - R_{i,t}\|_F + \|R_{i,t}\|_F \|S_{i,t+1} - S_{i,t}\|_F + \bar{R} \|S_{i,t+1} - S_{i,t}\| \right), \tag{81}$$

where (i) plugs in the expression in Equation (79); (ii) uses the fact that each element in $S_{i,t+1}$ has magnitude at most 1, and Cauchy-Schwartz inequality; (iii) comes from the definition of $\mathbb{I}, \mathbb{E}$ and $\bar{R}$; (iv) uses the definition of $U_{i,t}$; (v) is because triangle inequality; (vi) uses the fact that each element in $S_{i,t+1}$ has magnitude at most 1, and Cauchy-Schwartz inequality. Next, we aim to derive upper bound of $\|R_{i,t+1} - R_{i,t}\|_F$ in Equation (81).

$$
\begin{aligned}
\|R_{i,t+1} - R_{i,t}\|_F &= \left\| \big(\mathsf{MH}(M_{t+1}; X_i) - y_i\big) W^O (V'_{i,t+1}) - \big(\mathsf{MH}(M_t; X_i) - y_i\big) W^O (V'_{i,t}) \right\|_F \\
&= \left\| \big(\mathsf{MH}(M_{t+1}; X_i) - y_i\big) W^O (V'_{i,t+1}) - \big(\mathsf{MH}(M_t; X_i) - y_i\big) W^O (V'_{i,t+1}) + \right. \\
&\quad \left. \big(\mathsf{MH}(M_t; X_i) - y_i\big) W^O (V'_{i,t+1}) - \big(\mathsf{MH}(M_t; X_i) - y_i\big) W^O (V'_{i,t}) \right\|_F \\
&\overset{(i)}{\leq} \left\| \big(\mathsf{MH}(M_{t+1}; X_i) - \mathsf{MH}(M_t; X_i)\big) (V'_{i,t+1}) W^O \right\|_F + \left\| \big(\mathsf{MH}(M_t; X_i) - y_i\big)(V'_{i,t+1} - V'_{i,t}) W^O \right\|_F \\
&\overset{(ii)}{\leq} Z_i \|M_{t+1} - M_t\|_F \|\|X_i\|_F \sigma_{\max}(W^V) \|W^O\|_2 \\
&\quad + \big(\|\mathsf{MH}(M_{t+1}; X_i)\|_F + \|y_i\|_2\big) \|X_i\|_F \|W^V_{t+1} - W^V_t\|_F \|W^O\|_2 \\
&\overset{(iii)}{\leq} Z_i \|X_i\|_F \sigma_{\max}\big(W^V\big) \|W^O\|_2 \|M_{t+1} - M_t\|_F \\
&\quad + \big(n\sqrt{H}\sigma_{\max}\big(W^V\big) \|X_i\|_F \|W^O\|_2 + \|y_i\|_2\big) \|X_i\|_F \|W^V_{t+1} - W^V_t\|_F \|W^O\|_2 \\
&\overset{(iv)}{\leq} \big(Z_i \|X_i\|_F \sigma_{\max}\big(W^V\big) \|W^O\|_2 + (n\sqrt{H}\sigma_{\max}\big(W^V\big) \|X_i\|_F \|W^O\|_2 + \|y_i\|_2) \|X_i\|_F \|W^O\|_2\big) \\
&\quad \times \|M_{t+1} - M_t\|_F \\
&:= P_i \|M_{t+1} - M_t\|_F,
\end{aligned}
\tag{82}
$$

where (i) is because of the triangle inequality; (ii) uses the definition of $Z_i$ in Equation (74), Cauchy-Schwartz inequality and triangle inequality; (iii) uses the Cauchy-Schwartz inequality; (iv) reorganizes the terms in (iii). Plug Equation (82) into Equation (81), we can finally derive the bound for $\|\nabla_{W^Q} f(M_{t+1}; X_i) - \nabla_{W^Q} f(M_{t+1}; X_i)\|_F$.

$$
\begin{aligned}
&\|\nabla_{W^Q} f(M_{t+1}; X_i) - \nabla_{W^Q} f(M_{t+1}; X_i)\|_F \\
&\overset{(i)}{\leq} n\sqrt{H} \big(\|R_{i,t+1} - R_{i,t}\|_F + \|R_{i,t}\|_F \|S_{i,t+1} - S_{i,t}\|_F + \bar{R} \|S_{i,t+1} - S_{i,t}\|\big) \\
&\overset{(ii)}{\leq} n\sqrt{H} P_i \|M_{t+1} - M_t\|_F + 2\bar{R} n\sqrt{H} \|S_{i,t+1} - S_{i,t}\|_F \\
&\overset{(iii)}{\leq} n\sqrt{H} P_i \|M_{t+1} - M_t\|_F + 2\bar{R} n\sqrt{H} \sqrt{\phi_i^2 + \psi_i^2} \|M_{t+1} - M_t\|_F \\
&:= L_i^Q \|M_{t+1} - M_t\|_F,
\end{aligned}
$$

where (i) is from Equation (81); (ii) uses the definition of $\bar{R}$ in Equation (80); (iii) comes from Lemma 3 (3). Since $W^Q$ and $W^K$ are symmetric in the Transormer structure, similarly, we can derive $L_i^K$.
**Step 2:** In this step, we aim to derive bound for $\|\nabla_{W^v} f(M_{t+1}; X_i) - \nabla_{W^v} f(M_t; X_i)\|_F$.

$$\|\nabla_{W^V} f(M_{t+1}; X_i) - \nabla_{W^V} f(M_t; X_i)\|_F$$

$$\overset{(i)}{=} \left\| B_{i,t+1}^\top \left(\mathsf{MH}\left(M_{t+1}; X_i\right) - y\right) \left(W^O\right)^\top - B_{i,t}^\top \left(\mathsf{MH}\left(M_t; X_i\right) - y_i\right) \left(W^O\right)^\top \right\|_F$$

$$\overset{(ii)}{\leq} \left\| B_{i,t+1}^\top \left(\mathsf{MH}\left(M_{t+1}; X_i\right) - y_i\right) \left(W^O\right)^\top - B_{i,t+1}^\top \left(\mathsf{MH}\left(M_t; X_i\right) - y_i\right) \left(W^O\right)^\top \right\|_F$$

$$+ \left\| B_{i,t+1}^\top \left(\mathsf{MH}\left(M_t; X_i\right) - y_i\right) \left(W^O\right)^\top - B_{i,t}^\top \left(\mathsf{MH}\left(M_t; X_i\right) - y_i\right) \left(W^O\right)^\top \right\|_F$$

$$\overset{(iii)}{\leq} \|B_{i,t+1}\|_F\| \|\mathsf{MH}\left(M_{t+1}; X_i\right) - \mathsf{MH}\left(M_t; X_i\right)\|_F \|W^O\|_2 + \|B_{i,t+1} - B_{i,t}\|_F \|\mathsf{MH}\left(M_t; X_i\right) - y_i\|_F \|W^O\|_2$$

$$\overset{(iv)}{\leq} n\sqrt{H}\|X_i\|_F\|W^O\|_2 Z_i\|M_{t+1} - M_t\|_F + \|S_{i,t+1} - S_{i,t}\|_F \|X_i\|_F\|W^O\|_2 \left(\|\mathsf{MH}\left(M_{t+1}; X_i\right)\|_F + \|y_i\|_2\right)$$

$$\overset{(v)}{\leq} \sqrt{\phi_i^2 + \psi_i^2} \|X_i\|_F \|W^O\|_2 \left(n\sqrt{H}\sigma_{\max}\left(W^V\right) \|X_i\|_F \|W^O\|_2 + \|y_i\|_2\right) \|M_{t+1} - M_t\|_F$$

$$+ n\sqrt{H} \|W^O\|_2 \|X_i\|_F Z_i \|M_{t+1} - M_t\|_F$$

$$\overset{(vi)}{\leq} \left(\sqrt{\phi_i^2 + \psi_i^2} \|X_i\|_F \|W^O\|_2 \left(n\sqrt{H}\sigma_{\max}\left(W^V\right) \|X_i\|_F \|W^O\|_2 + \|y_i\|_2\right) + n\sqrt{H} \|W^O\|_2 Z_i\right) \|M_{t+1} - M_t\|_F$$

$$:= L_i^V \|M_{t+1} - M_t\|_F$$

where (i) is from Lemma 1 (1); (ii) uses triangle inequality; (iii) uses Cauchy-Schwartz inequality; (iv) comes from the definition of $B_{i,t,}$, $Z_i$(in Equation (74)), Cauchy-Schwartz inequality and triangle inequality; (v) comes from Lemma 2 (3) and Cauchy-Schwartz inequality; (vi) reorganizes (v).

Now we combine the result in **Step 1** and **Step 2**, and plug into Equation (78), we can finally derive

$$\|\nabla_W f\left(M_{t+1}; X\right) - \nabla_W f\left(M_t; X\right)\|_F \leq \sum_{i=1}^N (L_i^Q + L_i^K + L_i^V)\|M_{t+1} - M_t\|_F$$

$$\leq N(\max_i L_i^Q + \max_i L_i^K + \max_i L_i^V)\|M_{t+1} - M_t\|_F \tag{83}$$

$$:= G\|M_{t+1} - M_t\|_F. \tag{84}$$

$$\square$$

## 1.7 Proof of Lemma in Section 1.4

*Proof.* **Proof of Lemma 6 (1)**: We consider the differential of the element in the $k$-th row and $j$-th column. First, let us write down the closed form of each element:

$$(C_{ih})_{kj} = -\|X_{ik\cdot}W_h^Q - X_{ij\cdot}W_h^K\|^2/2\sqrt{d}$$

Next, we consider the differential of each element over $W_h^Q$:

$$d\left(C_{ih}\right)_{kj} = -\frac{1}{2\sqrt{d}} \left(\left\|X_{ik\cdot}\left(W_h^Q + d(W_h^Q)\right) - X_{ij\cdot}W_h^K\right\|^2 - \left\|X_{ik\cdot}\left(W_h^Q\right) - X_{ij\cdot}W_h^K\right\|^2\right)$$

$$= -\frac{1}{\sqrt{d}}\langle X_{ik\cdot}d(W_h^Q), X_{ik\cdot}W_h^Q - X_{ij\cdot}W_h^K\rangle + o\left(d(W_h^Q)\right),$$

where $o(d(W_h^Q))$ denotes the higher order of $d(W_h^Q)$. Leave out the higher order differential term, we derive

$$\|d\left(C_{ih}\right)_{kj}\|_F \leq \frac{1}{\sqrt{d}} \left(\|X_{ik\cdot}\|_2\|d(W_h^Q)\|_F \cdot \sigma_{\max}(W_h^Q)\|X_{ik\cdot}\|_2 + \|d(W_h^Q)\|_F \cdot \sigma_{\max}(W_h^K)\|X_{ik\cdot}\|_2\|X_{ij\cdot}\|_2\right)$$

$$\leq \frac{1}{\sqrt{d}}\|X_{ik\cdot}\|_2\|d(W_h^Q)\|_F(\sigma_{\max}(W_h^Q)\|X_{ik\cdot}\|_2 + \sigma_{\max}(W_h^K)\|X_{ij\cdot}\|_2)$$

$$\leq \frac{1}{\sqrt{d}}\|X_{ik\cdot}\|_2\sqrt{\sigma_{\max}^2(W_h^Q) + \sigma_{\max}^2(W_h^K)} \cdot \sqrt{\|X_{ik\cdot}\|_2^2 + \|X_{ij\cdot}\|_2^2}\|d(W_h^Q)\|_F$$

$$\|d\left(C_{ih}\right)\|_F = \sum_{k=1}^{n}\sum_{j=1}^{n}\|d(C_{ih})_{kj}\|_F^2$$

$$\leq \frac{1}{\sqrt{d}}\sum_{k=1}^{n}\sum_{j=1}^{n}\|X_{ik\cdot}\|_2\sqrt{\sigma_{\max}^2\left(W_h^Q\right)+\sigma_{\max}^2\left(W_h^K\right)}\cdot\sqrt{\|X_{ik\cdot}\|_2^2+\|X_{ij\cdot}\|_2^2}\|d(W_h^Q)\|_F$$

$$\leq \frac{1}{\sqrt{d}}\sqrt{\sigma_{\max}^2\left(W_h^Q\right)+\sigma_{\max}^2\left(W_h^K\right)}\sum_{k=1}^{n}\|X_{ik\cdot}\|\sqrt{n\|X_{ik\cdot}\|_2^2+\sum_{j=1}^{n}\|X_{ij\cdot}\|_F^2}\|d(W_h^Q)\|_F$$

$$\leq \frac{1}{\sqrt{d}}\sqrt{\sigma_{\max}^2\left(W_h^Q\right)+\sigma_{\max}^2\left(W_h^K\right)}\cdot\sqrt{\sum_{k=1}^{n}\|X_{ik\cdot}\|_F^2}\cdot\sqrt{\sum_{k=1}^{n}(n\|X_{ik\cdot}\|_2^2+\sum_{j=1}^{n}\|X_{ij\cdot}\|_F^2)}\|d(W_h^Q)\|_F$$

$$= \frac{1}{\sqrt{d}}\sqrt{\sigma_{\max}^2\left(W_h^Q\right)+\sigma_{\max}^2\left(W_h^K\right)}\cdot\|X_i\|_F\cdot\sqrt{2n}\|X_i\|_F\|d(W_h^Q)\|_F$$

$$= \sqrt{\frac{2n}{d}}\|X_i\|_F^2\sqrt{\sigma_{\max}^2\left(W_h^Q\right)+\sigma_{\max}^2\left(W_h^K\right)}\|d(W_h^Q)\|_F$$

$\square$

*Proof.* **Proof of Lemma 7 (2)**: First, let us write down the closed form of $\frac{\partial(C_{ih})_{kj}}{\partial W_h^Q}$. We have

$$\frac{\partial(C_{ih})_{kj}}{\partial W_h^Q} = -(X_{ik\cdot}W_h^Q - X_{ij\cdot}W_h^K)\mathbb{I}_d \otimes X_{ik\cdot} \tag{85}$$

Thus, we can derive upper bound for $\left\|d\left(\frac{\partial(C_{ih})_{kj}}{\partial W_h^Q}\right)\right\|_F$:

$$\left\|d\left(\frac{\partial\left(C_{ih}\right)_{kj}}{\partial W_h^Q}\right)\right\|_F = \left\|-\left(X_{ik\cdot}(W_h^Q+d(W_h^Q))-X_{ij\cdot}W_h^K\right)\mathbb{I}_d\otimes X_{ik}+\left(X_{ik\cdot}W_h^Q-X_{ij\cdot}W_h^K\right)\mathbb{I}_d\otimes X_{ik\cdot}\right\|_F/\sqrt{d}$$

$$= \|X_{ik\cdot}d(W_h^Q)\mathbb{I}_d\otimes X_{ik\cdot}\|_F/\sqrt{d}$$

$$\leq \|X_{ik\cdot}\|_2^2\|\mathbb{I}_d\|_F\|d(W_h^Q)\|_F/\sqrt{d}$$

$$= \|X_{ik\cdot}\|_2^2\|d(W_h^Q)\|_F \tag{86}$$

Thus, we have the following:

$$\left\|d\left(\frac{\partial\left(C_{ih}\right)}{\partial W_h^Q}\right)\right\|_F \leq \sum_{k=1}^{n}\sum_{j=1}^{n}\left\|d\left(\frac{\partial\left(C_{ih}\right)_{kj}}{\partial W_h^Q}\right)\right\|_F$$

$$\leq \|d(W_h^Q)\|_F\sum_{k=1}^{n}\sum_{j=1}^{n}\|X_{ik\cdot}\|_2^2$$

$$\leq n\|X_i\|_F^2\|d(W_h^Q)\|_F$$

$\square$

*Proof.* **Proof of Lemma 7 (3)**:

$$\left\|\frac{\partial f\left(M;X_i\right)}{\partial C_i}\right\|_F = \left\|\left((\mathsf{MH}\left(M;X_i\right)-y_i)\left(W^O\right)^\top\left(V_i'\right)^\top\right)\odot S_i\right\|_F$$

$$\geq \left\|\left((\mathsf{MH}\left(M;X_i\right)-y_i)\left(W^O\right)^\top\left(V_i'\right)^\top\right)\right\|_F\cdot\min|S_i|$$

$$\geq \min|V_i'W^O|\cdot\min|S_i|\cdot\|\mathsf{MH}\left(M;X_i\right)-y_i\|_2.$$

$\square$

*Proof.* **Proof of Lemma 7 (4)**:

$$
\left\| \frac{\partial f(M; X_i)}{\partial W_h^Q} \right\|_F = \left\| \mathrm{vec}\left( \frac{\partial f(M; X_i)}{\partial W_h^Q} \right) \right\|_2 = \left\| \mathrm{vec}\left( \frac{\partial f(M; X_i)}{\partial C_i} \right) \cdot \frac{\partial C_i}{\partial W_h^Q} \right\|_2
$$

$$
\leq \left\| \frac{\partial f(M; X_i)}{\partial C_i} \right\|_F \cdot \left\| \frac{\partial C_i}{\partial W_h^Q} \right\|_2
$$

$$
= \left\| \left( (\mathsf{MH}(M; X_i) - y_i)\left(W^O\right)^\top \left(V_i'\right)^\top \right) \odot S_i \right\|_F \cdot \sqrt{\frac{2n}{d}} \|X_i\|_F^2 \sqrt{\sigma_{\max}^2\left(W_h^Q\right) + \sigma_{\max}^2\left(W_h^K\right)}
$$

$$
\leq \sqrt{\frac{2n}{d}} \|X_i\|_F^3 \|W^O\|_2 \sigma_{\max}(W^V) \sqrt{\sigma_{\max}^2\left(W_h^Q\right) + \sigma_{\max}^2\left(W_h^K\right)} \|\mathsf{MH}(M; X_i) - y_i\|_2
$$

$\square$

*Proof.* **Proof of Lemma 8** (1): The proof is similar to the proof of Lemma 3 (1). So we do not include the details here. We can similarly derive

$$
\|f(M_{t+1}; X) - f(M_t; X)\|_2 \leq N \sqrt{\max_i Q_i'^2 + \max_i K_i'^2 + n^2 H \max_i \|X_i\|_F^2 \|W^O\|_2^2} \|M_{t+1} - M_t\|_F
$$

$$
:= Z' \|M_{t+1} - M_t\|_F
$$

$$(87)$$

$\square$

*Proof.* **Proof of Lemma 8 (2)**: By triangle inequality, we have

$$
\|\nabla_W f(M_{t+1}; X) - \nabla_W f(M_t; X)\|_F
$$
$$
\leq \|\nabla_{W^Q} f(M_{t+1}; X) - \nabla_{W^Q} f(M_{t+1}; X)\|_F + \|\nabla_{W^K} f(M_{t+1}; X) - \nabla_{W^K} f(M_{t+1}; X)\|_F
$$
$$
+ \|\nabla_{W^V} f(M_{t+1}; X) - \nabla_{W^V} f(M_{t+1}; X)\|_F
$$
$$
\leq \sum_{i=1}^N \big( \|\nabla_{W^Q} f(M_{t+1}; X) - \nabla_{W^Q} f(M_{t+1}; X)\|_F + \|\nabla_{W^K} f(M_{t+1}; X) - \nabla_{W^K} f(M_{t+1}; X)\|_F
$$
$$
+ \|\nabla_{W^v} f(M_{t+1}; X) - \nabla_{W^v} f(M_{t+1}; X)\|_F \big)
$$

$$(88)$$

**Step 1:** Derive upper bound for

$$
\|\nabla_{W^Q} f(M_{t+1}; X_i)) - \nabla_{W^Q} f(M_t; X_i))\|_F = \|\mathrm{vec}(\nabla_{W^Q} f(M_{t+1}; X_i)) - \mathrm{vec}(\nabla_{W^Q} f(M_t; X_i))\|_2.
$$

First, we give the vectorized expression of $\nabla_{W^Q} f(M_t; X_i)$. Recall we denote $U_i = \left( (\mathsf{MH}(M; X_i) - y_i)\left(W^O\right)^\top \left(V_i'\right)^\top \right) \odot S_i$. By Lemma 6, we can derive the close form of $\mathrm{vec}(\nabla_{W^Q} f(M_t; X_i))$:

$$
\mathrm{vec}(\nabla_{W^Q} f(M; X_i)) \overset{(i)}{=} \mathrm{vec}(U_i) \cdot \mathrm{vec}\left( \frac{\partial C_i}{\partial W_h^Q} \right)
$$

$$(89)$$

Further, recall we have defined $\bar{R}$ and the following inequality holds:

$$
\|U_i\|_F \leq \bar{R}
$$

$$(90)$$

Next, let us derive upper bound for $\|\nabla_{W^Q} f(M_{t+1}; X_i) - \nabla_{W^Q} f(M_{t+1}; X_i)\|_F$.

$$\|\nabla_{W^Q} f(M_{t+1}; X_i) - \nabla_{W^Q} f(M_{t+1}; X_i)\|_F$$

$$\overset{(i)}{=} \left\| \text{vec}(U_{i,t+1}) \cdot \left( \frac{\partial C_i(M_{t+1})}{\partial W_h^Q} \right) - \text{vec}(U_{i,t}) \cdot \left( \frac{\partial C_i(M_t)}{\partial W_h^Q} \right) \right\|_F$$

$$= \left\| \text{vec}(U_{i,t+1}) \cdot \left( \frac{\partial C_i(M_{t+1})}{\partial W_h^Q} \right) - \text{vec}(U_{i,t+1}) \cdot \left( \frac{\partial C_i(M_t)}{\partial W_h^Q} \right) \right. \tag{91}$$

$$\left. + \text{vec}(U_{i,t+1}) \cdot \left( \frac{\partial C_i(M_t)}{\partial W_h^Q} \right) - \text{vec}(U_{i,t}) \cdot \left( \frac{\partial C_i(M_t)}{\partial W_h^Q} \right) \right\|_F$$

$$\overset{(ii)}{\leq} \| \text{vec}(U_{i,t+1})\|_2 \left\| \frac{\partial C_i(M_{t+1})}{\partial W_h^Q} - \frac{\partial C_i(M_t)}{\partial W_h^Q} \right\|_2 + \| \text{vec}(U_{i,t+1} - U_{i,t})\|_2 \left\| \frac{\partial C_i(M_t)}{\partial W_h^Q} \right\|_2$$

$$\overset{(iii)}{\leq} \bar{R}\sqrt{n} \|X_i\|_F^2 \cdot \left\| d\left(W_h^Q\right) \right\|_F + \|\text{vec}(U_{i,t+1} - U_{i,t})\|_F \cdot \sqrt{n} \|X_i\|_F^2 \cdot \left( \sigma_{\max}\left(W_h^Q\right) + \sigma_{\max}\left(W_h^K\right) \right)$$

$$\leq \bar{R}\sqrt{n} \|X_i\|_F^2 \cdot \left\| d\left(W_h^Q\right) \right\|_F + \sqrt{n} \|X_i\|_F^2 \cdot \left( \sigma_{\max}\left(W_h^Q\right) + \sigma_{\max}\left(W_h^K\right) \right) \|R_{i,t+1} \odot S_{i,t+1} - R_{i,t} \odot S_{i,t}\|_F$$

$$\leq \bar{R}\sqrt{n} \|X_i\|_F^2 \cdot \left\| d\left(W_h^Q\right) \right\|_F + \sqrt{n} \|X_i\|_F^2 \cdot \left( \sigma_{\max}\left(W_h^Q\right) + \sigma_{\max}\left(W_h^K\right) \right)$$

$$\times \left( \|(R_{i,t+1} - R_{i,t}) \odot S_{i,t+1}\|_F + \|R_{i,t} \odot S_{i,t+1} - R_{i,t} \odot S_t)\|_F \right)$$

$$\leq \bar{R}\sqrt{n} \|X_i\|_F^2 \cdot \left\| d\left(W_h^Q\right) \right\|_F + \sqrt{n} \|X_i\|_F^2 \cdot \left( \sigma_{\max}\left(W_h^Q\right) + \sigma_{\max}\left(W_h^K\right) \right)$$

$$\times \left( \|R_{i,t+1} - R_{i,t}\|_F + \|R_{i,t}\|_F \|S_{i,t+1} - S_{i,t}\|_F \right) \tag{92}$$

Next, we aim to derive upper bound of $\|R_{i,t+1} - R_{i,t}\|_F$ in Equation (92). Similar to the derivation in Equation (81), we can derive

$$\|R_{i,t+1} - R_{i,t}\|_F \overset{(iv)}{\leq} \left( Z_i' \|X_i\|_F \sigma_{\max}\left(W^V\right) \|W^O\|_2 + (n\sqrt{H}\sigma_{\max}\left(W^V\right) \|X_i\|_F \|W^O\|_2 + \|y_i\|_2) \|X_i\|_F \|W^O\|_2 \right)$$

$$\times \|M_{t+1} - M_t\|_F$$

$$:= P_i' \|M_{t+1} - M_t\|_F, \tag{93}$$

Plug Equation (82) into Equation (92), we can finally derive the bound for $\|\nabla_{W^Q} f(M_{t+1}; X_i) - \nabla_{W^Q} f(M_{t+1}; X_i)\|_F$.

$$\|\nabla_{W^Q} f(M_{t+1}; X_i) - \nabla_{W^Q} f(M_{t+1}; X_i)\|_F$$

$$\leq \bar{R}\sqrt{n} \|X_i\|_F^2 \cdot \left\| d\left(W_h^Q\right) \right\|_F + \sqrt{n} \|X_i\|_F^2 \cdot \left( \sigma_{\max}\left(W_h^Q\right) + \sigma_{\max}\left(W_h^K\right) \right)$$

$$\times \left( \|R_{i,t+1} - R_{i,t}\|_F + \|R_{i,t}\|_F \|S_{i,t+1} - S_{i,t}\|_F \right)$$

$$\leq \bar{R}\sqrt{n} \|X_i\|_F^2 \cdot \|M_{t+1} - M_t\|_F + \sqrt{n} \|X_i\|_F^2 \cdot \left( \sigma_{\max}\left(W_h^Q\right) + \sigma_{\max}\left(W_h^K\right) \right)$$

$$\times \left( P_i' \|M_{t+1} - M_t\|_F + \sqrt{n}\bar{R} \|X_i\|_F^2 \cdot \left( \sigma_{\max}\left(W_h^Q\right) + \sigma_{\max}\left(W_h^K\right) \right) \|M_{t+1} - M_t\|_F \right)$$

$$:= L_i^{Q'} \|M_{t+1} - M_t\|_F,$$

and plug into Equation (78), we can finally derive

$$\|\nabla_W f(M_{t+1}; X) - \nabla_W f(M_t; X)\|_F \leq \sum_{i=1}^{N} (L_i^Q + L_i^K + L_i^V) \|M_{t+1} - M_t\|_F$$

$$\leq N(\max_i L_i^Q + \max_i L_i^K + \max_i L_i^V) \|M_{t+1} - M_t\|_F \tag{94}$$

$$:= G \|M_{t+1} - M_t\|_F. \tag{95}$$

$\square$

