# OpenReview forum: "Unraveling the Gradient Descent Dynamics of Transformers"
_NeurIPS.cc/2024/Conference — NeurIPS 2024 poster_

### Official Review · Reviewer_ynwg · 2024-07-11

**Soundness:** 3
**Presentation:** 3
**Contribution:** 2
**Rating:** 5
**Confidence:** 3

**Summary:**

This paper proves that with appropriate initialization GD can train a Transformer model with either Softmax or Gaussian kernel to achieve a global optimal solution. Besides, this paper highlights the Gaussian attention kernel exhibits much favorable behavior than Softmax in certain scenarios.

**Strengths:**

1. This paper gives a theoretical analysis of training dynamics of Transformers, and provides conditions to guarantee global convergence.

2. The comparison between Gaussian and Softmax kernels is interesting.

**Weaknesses:**

1. It is unclear what new insights could be gained from the global convergence result Theorem 2, given that Wu et al [2024] already have one. The weight $W^O$ is trainable in Wu et al [2024], which is in line with practice, but $W^O$ is fixed in the present paper.

2. The comparison between the Gaussian and Softmax kernels only updates $W^Q$.

**Questions:**

1. The conditions in Theorem 2 is different from those in Wu et al [2024]. How do these conditions compare with each other? Which one is weaker? What new insights can we obtain?

2. The authors emphasize that one important feature of the present work is that they delve in the roles of individual variables. Why is this important? Can the results obtained with one variable trainable and others fixed explain their roles when they are simultaneously updated?

3. On line 286, is it implicitly assumed that the global optimum should have zero loss? Why must this be true?

4. Lines 376-377 claim that the results in this paper explain why transformers perform exceptionally well. Can you clarify the explanation? How does it follow from the results of the paper?

5. There are several sentences like "only updating $W^V$ already leads to global convergence". Are they suggesting that it is easier to optimize more variables? How should we understand these sentences?

**Limitations:**

Not quite. See the weakness section.

---

> ### Author Rebuttal · Authors · 2024-08-06
>
> >>**Your comment**: The new insight we can gain compared to Wu et al [2024]; the fixed $W^O$ in optimization.
>
> **Response**: See the general rebuttal at the top of the page **Comment 1**.
>
> >> **Your comment**: The comparison between the Gaussian and Softmax kernels only updates $W^Q$.
>
> **Response**: We'd like to clarify that we also studied the role of the variable $W^V$ in Gaussian and Softmax kernel and found them to be the same. We have commented on the similarity between the two Transformers in **Remark 3**. Specifically, the conclusions in Theorem 1 and Theorem 2 hold for Transformers with Gaussian attention, and the linear decrease of the loss function also comes from $W^V$ when $W^Q,W^K,W^V$ are updated.  Since the conclusion and proof technique are very similar to Theorem 1 and Theorem 2, we did not include the details in the paper. To avoid confusion, we will add a corollary to clearly state the similarity of $W^V$ in both Transformers in our revised version.
>
> >>**Your question**: The comparison of assumptions in Wu et al [2024] and our paper.
>
> **Response**:  See the general rebuttal at the top of the page **Comment 2**.
>
> >>Your question regarding the importance of analyzing individual variables. Can the results obtained with one variable trainable and others fixed explain their roles when they are simultaneously updated?
>
> **Response**:
> **The importance of unravelling individual variables**: First, unravelling the individual role of each variable helps us understand the optimization bottlenecks in Transformers [2,3]. For example, in [2], it is shown that the rank collapse of the attention kernel can occur when the variable $W^Q,W^K$ are updated, which can lead to failure in optimization. Second, acceleration optimization algorithms can be designed based on the understanding of the individual role of each variable [3,5]. In these works, Nyström method is used in gradient computation to greatly accelerate the training. The algorithm design is specific to the training dynamics of individual variables in Transformers. Third, analyzing the training dynamics of individual variables leads to a new network design that improves the optimization performance [3,4,5]. In these works, attention structures other than Softmax attention are proposed to improve the generalization and computational efficiency.
>
> **Simultaneously update**: Yes, the property obtained when analyzing a single variable updates does carry over to the situation when this variable is updated to other variables simultaneously. The reason is as follows: Our over-parameterization structure and initialization ensure all the variables **always stay in a locally near-convex region**. Within this region, the optimization landscape is relatively smooth, and the partial gradient of any single variable does not change much when other variables are updated, thus analyzing the gradients of different variables separately is able to explain the role of each variable when they are updating simultaneously.
> >>Your question regarding the existence of global optimum in line 286.
>
> **Response**: Yes, it is implicitly assumed that the global optimum solution has zero loss. This is true because of the over-parameterization of the network, where the network size is $\Omega(Nn)$ ($N$ is the sample size). When there are more parameters than samples, there exists a solution that can completely fit all the samples. Further, in some other works with similar settings, the existence of global optimal solution can also be derived from the limit of a  Cauchy sequence (See [8] Appendix B.7). To make the proof more rigorous, we will add a similar derivation of the Cauchy sequence in our revised version.
>
> >>Your question regarding the statement "transformers perform exceptionally well".
>
> **Response**: We claim that our result explains why Transformers perform exceptionally well because we derive a **global convergence** analysis of training Transformers (Theorems 1 and 2). These two theorems show that even the plain gradient decent algorithm can find the global optimal solution with a network size of $\Omega(Nn)$, allowing the training loss to decrease to $0$. This provides theoretical proof for the outstanding training performance of Transformers.
> >>Your question regarding the sentences like "only updating $W^V$ already leads to global convergence".
>
> **Response**: No, it does not suggest that it is easier to optimize more variables. As we aim to scrutinize the role of each variable in the attention kernel, we found that the global convergence is dominated by $W^V$, which prompted us to later focus on studying the **role of $W^Q$ and $W^K$** in optimization. These sentences are meant to emphasize that the role of $W^Q, W^K$ remains unclear if updating $W^V$ alone already leads to global convergence (See Theorem 1,2), i.e, does updating $W^Q,W^K$ contribute to the linear convergence of loss function? Thus, these statements lead to our discussion on $W^Q$ alone in Theorem 3.
>
> [1] Wu, Yongtao, et al. "On the convergence of encoder-only shallow transformers." Advances in Neural Information Processing Systems 36 (2024).
>
> [2] Noci, Lorenzo, et al. "Signal propagation in transformers: Theoretical perspectives and the role of rank collapse." Advances in Neural Information Processing Systems 35 (2022): 27198-27211.
>
> [3] Chen, Yifan, et al. "Skyformer: Remodel self-attention with gaussian kernel and nystr\" om method." Advances in Neural Information Processing Systems 34 (2021): 2122-2135.
>
> [4] Choromanski, Krzysztof, et al. "Rethinking attention with performers." arXiv preprint arXiv:2009.14794 (2020).
>
> [5] Lu, Jiachen, et al. "Soft: Softmax-free transformer with linear complexity." Advances in Neural Information Processing Systems 34 (2021): 21297-21309.
>
> [6] Nguyen, Quynh N., and Marco Mondelli. "Global convergence of deep networks with one wide layer followed by pyramidal topology." Advances in Neural Information Processing Systems 33 (2020): 11961-11972.

---

> > ### Comment · Reviewer_ynwg · 2024-08-08
> >
> > Thanks for the response. The response claims global convergence on the one hand, and initialization in a locally near convex region. How do you reconcile these two claims?

---

> > > ### Author Response · Authors · 2024-08-09
> > > **Response to Reviewer ynwg's question**
> > >
> > > First, let us clarify that global convergence in our theorems means **convergence to global optimal solution**. The definition is also used in other paper that analyzes the optimization of networks, e.g,[1]. And this definition is different from the traditional definition of "global convergence", which means that an algorithm converges (to certain solution) regardless of initialization. To avoid confusion, we will correct it to "convergence to global optimal solution" in our revised version.
> > >
> > > Second, with the clarification above, we need to clarify that the condtion of initialization in near-convex condition **does not conflict** with our analysis of achieving global optimal solution in Theorem 1,2 and 3. Instead, it is an **essential condition** for deriving the result in our paper. This type of claim (local initialization to achieve global optimal solution) has been common [1,2,3], though in some works it is not pointed out directly.
> > >
> > >
> > >
> > > [1] Nguyen, Quynh N., and Marco Mondelli. "Global convergence of deep networks with one wide layer followed by pyramidal topology." Advances in Neural Information Processing Systems 33 (2020): 11961-11972.
> > >
> > > [2] Wu, Yongtao, et al. "On the convergence of encoder-only shallow transformers." Advances in Neural Information Processing Systems 36 (2024).
> > >
> > > [3] Song, Bingqing, et al. "Fedavg converges to zero training loss linearly for overparameterized multi-layer neural networks." International Conference on Machine Learning. PMLR, 2023.

---

> > > > ### Comment · Reviewer_ynwg · 2024-08-10
> > > >
> > > > Thanks for the clarification. I'll keep my score.

---

### Official Review · Reviewer_mp2C · 2024-07-12

**Soundness:** 3
**Presentation:** 3
**Contribution:** 3
**Rating:** 6
**Confidence:** 2

**Summary:**

The authors establish different convergence theorems on the training of a single layer transformer with different trainable weight matrices and kernel functions. They prove that, under certain conditions, a one-layer Transformer with a Gaussian kernel converges faster than one with a Softmax kernel. Finally, they conduct empirical experiments to show the effectiveness of Gaussian kernel compared with Softmax kernel, validating their theory.

**Strengths:**

The paper is overall clearly written. The authors delve into the role of different weight matrices in Transformer optimization and establish different convergence theorems with regard to different trainable weight matrices. This aspect is novel. Their theory also suggests that Gaussian kernel may be more effective than the widely used softmax kernel, which if validated in larger models, can have a great impact on the field.

**Weaknesses:**

Minor points:
1.	It may be beneficial to include insights in the main text about the critical steps in the proof that result in the different convergence rates of the Gaussian and Softmax kernels in Theorem 3.

Major points:
1.	The experimental results in Figure 2 suggest that the Transformer with a Gaussian kernel converges faster. However, the difference in convergence seems more like a difference in the constant factor rather than a difference in the convergence rate.

**Questions:**

1.	The authors construct a compelling example to illustrate the failure of a Transformer with a Gaussian kernel, but it requires a specific initialization form. To achieve a similar convergence rate for the Softmax kernel, how many additional conditions are required?
2.	I don’t fully understand why optimizing the query matrix alone leads to similar dynamics as when optimizing both the query and the key matrix. Could you clarify this?

**Limitations:**

The author has discussed some of the limitations in the paper. Specifically, the authors explain why they choose to study a simple one-layer transformer with regression loss while in practice, transformer models are usually trained with cross-entropy loss and with multiple layers.

---

> ### Author Rebuttal · Authors · 2024-08-06
>
> >>**Your comment**:  The difference in convergence speed in Fig. 2 seems like constant level with same rate.
>
> **Response**: Thank the reviewer for the comment. We need to clarify that it is **reasonable** that the difference in convergence rate in Fig. 2 is constant level. Further, we provide **empirical reason** for the different theoretical convergence rates in Theorem 3 by landscape visualization in Fig. 4.
>
> Frist, Transformers with Gaussian and Softmax attentions theoretically should have **the same convergence rate** in our experiment setting in Fig. 2. Notice that the algorithm we use in Fig. 2 is updating **all the variables** in the attention ($W^Q,W^K,W^V$). As discussed in Remark 3, when all the variables are updated, both Transformers could achieve linear convergence rate with appropriate network size and initialization. Thus, the theoretical difference in convergence rate is constant level. However, due to the bottleneck of optimizing $W^Q$ (or $W^K$) in Softmax Transformer as illustrated in Theorem 3, Gaussian Transformer still converges **faster** than Softmax Transformer when $W^Q, W^K$ are included in optimization.
>
> Second, in terms of the different theoretical convergence rate in Theorem 3 when **only $W^Q$ (or $W^K$)** is updated in both Transformers, we  provide **empirical reason** for the different convergence rates in Theorem 3 by landscape visualization in Fig. 4. By showing the existence of **more local solutions** in Transformers with Softmax attention, we provide evidence for the comparison of the convergence rates: Compared to Transformer with Gaussian attention, a Transformer with Softmax attention is more likely to **obtain a sublinear convergence to stationary points**, not global optimum.
>
>
> >>**Your comment**: The insights of critical proof steps that cause different convergence rates in Theorem 3.
>
> **Response**: We thank the reviewer for the suggestion. We will add the proof sketch of Theorem 3 in the revised version. Here we will briefly summarize the critical steps.
>
> **Step 1**: Derive the **closed form gradient** of loss function over variable $W^Q$. Please see Lemma 1 equation (23),(24) in line 778 for Softmax attention Transformer, and Lemma 6 equation (41), (42) in line 821 for Gaussian attention Transformer. Intuitively, the gradient of Softmax attention is much more complicated than Gaussian attention Transformer, which will lead to a more complicated landscape and more local solutions.
>
> **Step 2**: **Analyze the gradient** of Transformers with both kernels with the **same initialization** Equation (12). For Gaussian attention Transformer, it can be iteratively shown during the gradient descent training: 1) The variable $W^Q$ is bounded; 2) The PL condition holds (i.e, the optimization landscape remains near-convex); 3) The loss function decreases linearly. (Please see Equation (52) in line 829.) For the Softmax attention Transformer, there is no guarantee that the PL condition holds during iterative gradient descent update. To illustrate this claim, we construct a concrete counterexample to show that the PL condition does not hold. (Please see line 284).
>
> >>**Your question**: The authors construct a compelling example to illustrate the failure of a Transformer with a Gaussian kernel, but it requires a specific initialization form. To achieve a similar convergence rate for the Softmax kernel, how many additional conditions are required?
>
> **Response**: Respectfully, we need to clarify that we **do not** construct an example to illustrate the failure of the Transformer with a **Gaussian kernel**. Instead, we provide a detailed example to illustrate that the Transformer with Softmax attention converges to a **stationary point** with a sublinear rate, but fails to achieve linear global convergence. (Please see Equation (14) in Theorem 2 and the failure to satisfy the PL condition in lines 284 and 285).
>
> **Additional condition for Softmax Transformer**: For a Transformer with Softmax attention, the **PL condition or convexity of loss function** should be assumed during the **whole training phase** to achieve the same linear convergence rate as a Transformer with Gaussian attention. This condition is much more stringent than the condition for Gaussian attention, where the PL condition is assumed **only at initialization**.
> >>**Your question**: I don’t fully understand why optimizing the query matrix alone leads to similar dynamics as when optimizing both the query and the key matrix. Could you clarify this?
>
> **Response**: The similarity comes from the symmetry of $W^Q$ and $W^K$ in attention mechanism. The symmetry means the training dynamics/gradients of  $W^Q$ and $W^K$ have **exactly the same** properties. Please refer to Appendix 1.2 Lemma 1 (3). We write down the closed form gradient of $W_h^Q$ over $f$ for the Softmax Transformer. Similarly, we can derive the gradient over $W_h^K$, with the only difference from Lemma 1 (3) is that, the $W_h^K$ term on the right side of the equation is replaced with $W_h^Q$. Thus, the gradients of $W^Q$ and $W^K$ have the **same** structure. As long as we initialize $W^Q$ and $W^K$ with the same properties, the training dynamics are also **the same** for both gradients. Intuitively, this stems from the fact that $W^Q$ and $W^K$ are in symmetric positions with respect to calculating their gradients in the attention head, see equation (3).
>
> [1] Huang, Baihe, et al. "Fl-ntk: A neural tangent kernel-based framework for federated learning analysis." International Conference on Machine Learning. PMLR, 2021.
>
> [2] Gao, Tianxiang, et al. "A global convergence theory for deep relu implicit networks via over-parameterization." arXiv preprint arXiv:2110.05645 (2021).
>
> [3] Wu, Yongtao, et al. "On the convergence of encoder-only shallow transformers." Advances in Neural Information Processing Systems 36 (2024).

---

> > ### Comment · Reviewer_mp2C · 2024-08-09
> >
> > I thank the authors for the response! The authors have clarified the new insights and novel contributions compared with previous works. As a result, I am raising my score to 6.

---

### Official Review · Reviewer_cxoC · 2024-07-13

**Soundness:** 3
**Presentation:** 1
**Contribution:** 2
**Rating:** 6
**Confidence:** 3

**Summary:**

This paper analyzes the convergence behavior of Transformer models with different attention mechanisms, specifically comparing Softmax and Gaussian kernel attention. The authors provide theoretical results on the conditions for global convergence and empirically demonstrate differences in optimization landscapes between the two attention types. The work aims to provide insights into why Transformers perform well and the potential advantages/disadvantages of Softmax attention.

**Strengths:**

- The analysis of Gaussian kernel attention is insightful and novel. The authors show theoretically and empirically that Gaussian attention can lead to better convergence properties compared to Softmax attention. This provides valuable understanding of alternative attention mechanisms.
- The paper addresses an important question about the optimization dynamics of Transformers, which is crucial given their widespread use. The motivation to understand why Transformers work well and potential limitations of - Softmax attention is timely and relevant.
- The theoretical analysis is reasonably thorough, with the authors deriving conditions for global convergence for both Softmax and Gaussian attention under different update scenarios (Theorems 1-3). This helps formalize the intuitions about attention behavior.

**Weaknesses:**

- The empirical evaluation is limited and doesn't fully validate the theoretical claims in practical settings. The experiments use simplified Transformer models on relatively small datasets (IMDB and Pathfinder). It would be more convincing to see results on larger, more complex Transformer architectures and standard NLP benchmarks.
- The paper's contribution relative to prior work is not clearly articulated. While the authors cite some related papers, they don't adequately explain how their results extend or differ from existing analyses of Transformer convergence, e.g., [1].
- The practical implications of the theoretical results are not sufficiently discussed. It's unclear how the insights about Gaussian attention could be applied to improve real-world Transformer models or training.

[1] Y. Huang, Y. Cheng, and Y. Liang. In-context convergence of transformers. arXiv preprint arXiv:2310.05249, 2023.

**Questions:**

See weakness above.

**Limitations:**

The authors acknowledge some limitations in Section 6, noting that they require strong initialization and a large embedding size to obtain global convergence guarantees. They also mention the gap between their analysis and real-world scenarios. However, a more thorough discussion of limitations would strengthen the paper.

---

> ### Author Rebuttal · Authors · 2024-08-06
>
> >>**Your comment**: The empirical evaluation is limited and doesn't fully validate the theoretical claims in practical settings. The experiments use simplified Transformer models on relatively small datasets (IMDB and Pathfinder). It would be more convincing to see results on larger, more complex Transformer architectures and standard NLP benchmarks.
>
> **Response**: Thanks the reviewer for the comment. First, we need to clarify that the goal of our paper is to investigate the **training dynamics** and **landscape** of Transformer model. The main purpose of our experiment is to **provide a concrete example** to illustrate the existence of local solution in Softmax attention. Similar experiments of **shallow** networks can be found in [1,2,9], which have settings similar to ours.
>
> We agree that additional experiments on larger models could potentially provide further support for our theory. However, due to computational budget constraints, we are unable to complete these experiments during the rebuttal period. We will add results on these larger models at a later time.
>
> >>**Your comment**: The paper's contribution relative to prior work is not clearly articulated. While the authors cite some related papers, they don't adequately explain how their results extend or differ from existing analyses of Transformer convergence, e.g., [4].
>
> **Response**: We thank the reviewer for the comment. We will add the following detailed comparison with the related works in the revised version:
> 1. Comparison with [3,4]: These works analyze the convergence of in-context training, where a prompt is contructed with all the training samples and a single test sample. The goal of these works is to achieve the zero test loss (in expectation) by optimizing over the loss function modeled by the propmpt. However, our analysis is based on standard empirical loss minimization, which does not involve any prompt construction.
> 2. Comparison with [5,6,7,8]: This line of works analyze the global convergence of training over-parameterized fully connected networks. However, Transformer structure and the optimization landscape in our study are more complicated.
> 3. Comparison with [9]: This work is closely related to our work. However, the primary goal of [9] and our work is **different**. In [9], the main purpose is to **build a convergence theory** of shallow Transformer with realistic structure and initialization, but they do not provide roles for different matrices in the convergence. While in our paper, we focus on **investigating the role of each variables** within the attention mechanism in optimization. Further, we analyze **the convergence behavior of different attention kernels**, i.e, Softmax, Gaussian, which is not covered by [9].
>
>
>
> >>**Your comment**: The practical implications of the theoretical results are not sufficiently discussed. It's unclear how the insights about Gaussian attention could be applied to improve real-world Transformer models or training.
>
> **Response**: We thank the reviewer for the comment. Here we discuss the insights about the benefits of using a Gaussian kernel in training Transformer models. As discussed in [1], Gaussian kernels have a natural interpretation of assigning “attention” to different tokens, similar to Softmax attention. However, the Gaussian kernel performs intrinsic normalization, which allows the attention mechanism to have a more reasonable condition number than Softmax attention. Therefore, designing specific Transformers with Gaussian attention can greatly improve the stability of model training. For example, in [1] Table 1, the kernelized attention (Gaussian kernel) and Skyformer (modified Gaussian kernel) show better performance than Softmax attention. In [10], it is showed that the Gaussian Transfomer is more efficient and generalizes better than Softmax Transfomer.
>
> [1] Chen, Yifan, et al. "Skyformer: Remodel self-attention with gaussian kernel and nystr" om method." Advances in Neural Information Processing Systems 34 (2021): 2122-2135.
>
> [2] Tay, Yi, et al. "Long range arena: A benchmark for efficient transformers." arXiv preprint arXiv:2011.04006 (2020).
>
> [3] Zhang, Ruiqi, Spencer Frei, and Peter L. Bartlett. "Trained transformers learn linear models in-context." arXiv preprint arXiv:2306.09927 (2023).
>
> [4] Huang, Yu, Yuan Cheng, and Yingbin Liang. "In-context convergence of transformers." arXiv preprint arXiv:2310.05249 (2023).
>
> [5] Allen-Zhu, Zeyuan, Yuanzhi Li, and Zhao Song. "A convergence theory for deep learning via over-parameterization." International conference on machine learning. PMLR, 2019.
>
> [6] Du, Simon, et al. "Gradient descent finds global minima of deep neural networks." International conference on machine learning. PMLR, 2019.
>
> [7] Nguyen, Quynh N., and Marco Mondelli. "Global convergence of deep networks with one wide layer followed by pyramidal topology." Advances in Neural Information Processing Systems 33 (2020): 11961-11972.
>
> [8] Allen-Zhu, Zeyuan, Yuanzhi Li, and Zhao Song. "On the convergence rate of training recurrent neural networks." Advances in neural information processing systems 32 (2019).
>
> [9] Wu, Yongtao, et al. "On the convergence of encoder-only shallow transformers." Advances in Neural Information Processing Systems 36 (2024).
>
> [10] Lu, Jiachen, et al. "Soft: Softmax-free transformer with linear complexity." Advances in Neural Information Processing Systems 34 (2021): 21297-21309.

---

> > ### Comment · Reviewer_cxoC · 2024-08-10
> > **Thank you. Raise score from 5 to 6. More questions.**
> >
> > Thank you for your rebuttal. It fixed some of my concerns and I raised my score from 5 to 6.
> >
> > On the other hand, I have some other further questions.
> > - In Remark 1, it seems that the only assumption for data and initialization is that $B_0$ is full rank. Then, by over-parameterization, i.e.,  $D = \Theta (N n)$, the loss landscape is nearly convex around initialization. This sounds very similar to NTK or a lazy learning regime. I wonder whether there are any feature learning or rich learning regime effects in your analysis.
> > - Is there any idea about the generalization with some assumptions on data distribution?
> > - Why Gaussian kernel is $-\\|XW_Q - XW_K\\|_2^2$ rather than $(XW_Q W_K^T X^T)^2$? From my perspective, $-\\|XW_Q - XW_K\\|_2^2 \approx C  + 2 XW_Q W_K^T X^T$ with some constant $C$, which is not far different from the standard kernel.

---

> > > ### Author Response · Authors · 2024-08-12
> > > **Reply to Reviewer cxoC's questions**
> > >
> > > We thank the reviewer for the feedback and raise in score!
> > >
> > > **Comment**: In Remark 1, it seems that the only assumption for data and initialization is that $B_0$ is full rank. Then, by over-parameterization, i.e. $D=\Omega(Nn)$, the loss landscape is nearly convex around initialization. This sounds very similar to NTK or a lazy learning regime. I wonder whether there are any feature learning or rich learning regime effects in your analysis.
> > >
> > > **Response**: Yes, our optimization analysis is similar to the NTK regime with lazy update. Our current theory does not consider feature learning or the rich learning effect. However, we believe it is possible to extend our analysis towards the feature learning regime. For example, the high-dimensional embedding can be further modeled by a function that extracts data features, e.g, networks, allowing the attention head to be interpreted as computing the "correlation" of extracted features. We plan to investigate the feature learning regime in our future work.
> > >
> > > **Comment**: Is there any idea about the generalization with some assumptions on data distribution?
> > >
> > > **Response**: Yes, generalization analysis can be derived from our theory with certain assumptions on data distribution and model weights, although it is not analyzed in this paper as we primarily focus on the optimization landscape. As a concrete example, it is possible to extend our analysis by following the framework in [1], where the data is balanced and a binary classification problem is considered. We can further derive a similar generalization bound as [1] Theorem 1.
> > >
> > >
> > > **Comment:** The formula of Gaussian kernel.
> > >
> > > **Response**: Respectfully, we would like to clarify that the formula of Gaussian kernel in your comment is missing an additional activation function. With different activations included, the Gaussian kernels (defined in [2, 3] and equation (11)) and Softmax kernels differ significantly, as the Softmax activation considers the entire row of the attention matrix, whereas the Gaussian does not.
> > > Specifically, the $k$-th row and $j$-th column of Gaussian attention is given by (see equation (11)):
> > > $$S\left(W\_h^Q, W_h^K ; X_i\right)\_{k j}=\operatorname{\exp}\left(-\frac{1}{\sqrt{d}}\left(X\_{ik\cdot} W_h^Q-X\_{ij\cdot} W\_h^K\right)^2\right)$$.
> > > The Softmax attention is given by:
> > > $$S\_{ih}:=\text{Softmax}\left(\frac{X\_{i} W_h^Q\left(X\_{i} W\_h^K\right)^{\top}}{\sqrt{d}} \in \mathbb{R}^{n\times n}\right).$$
> > > Notice that the entry in Gaussian kernel is only related to the $k$-th and $j$-th token in sample $X_i$. In contrast, in Softmax attention, each entry is also related to the entries in the same row, since the Softmax activation computes the regularization based on a row. This leads to a more complex optimization landscape and makes the convergence more challenging. These two different kernels also result in different performance on some tasks [2, 3].
> > >
> > >
> > > [1] Li, Hongkang, et al. "A theoretical understanding of shallow vision transformers: Learning, generalization, and sample complexity." arXiv preprint arXiv:2302.06015 (2023).
> > >
> > > [2] Lu, Jiachen, et al. "Soft: Softmax-free transformer with linear complexity." Advances in Neural Information Processing Systems 34 (2021): 21297-21309.
> > >
> > > [3] Chen, Yifan, et al. "Skyformer: Remodel self-attention with gaussian kernel and nystr" om method." Advances in Neural Information Processing Systems 34 (2021): 2122-2135.

---

> > > > ### Comment · Reviewer_cxoC · 2024-08-12
> > > > **Thank you and raise my confidence score from 2 to 3**
> > > >
> > > > Thank you for your reply. Now, I understand this paper better, and my confidence score has increased from 2 to 3.

---

### Author Rebuttal · Authors · 2024-08-06

We thank the reviewers for the comments and suggestions. We will summarize the strength of our work from reviewers as following:
1. The analysis of Transformer is under realistic setting, and requires no strict data assumption.
2. We delve into each variables within attention kernels. The analysis on Transformer landscape and comparison between different attention kernels are novel and significant.
4. The analysis is non-trivial and requires sufficient techniques.

Regarding the weakness and questions, we will address the several comments about the comparison with [1] here. Please see our detailed response individually for other comments. We are glad to answer your further questions.

**Comment 1**: The new insight of our Theorem 2 compared to [1]; the fixed $W^O$ in our paper.

**Response**: We thank the reviewer for the comment. We state the differences between [1] and our work in terms of optimization algorithm and discuss the new insight we can gain from our setting.

**Difference in optimization algorithm**: In [1], all the variables ($W^Q,W^K,W^V,W^O$) are updated to build a convergence theory, while in our work, we consider updating $W^Q,W^K,W^V$ in Theorem 2. We find that including $W^O$ in the analysis will **hide the true role** of $W^Q,W^K,W^V$ in optimization (see the explanation in the following paragraph). Given that we aim to unravel the role of different variables **within the attention head** in the convergence analysis, we leave out $W^O$. However, we need to point out that if we include $W^O$ as a variable, we can still achieve the same convergence rate as in Theorem 2.

**Difference in insight**: In Proposition 1 of [1], the linear decrease of the loss function **only** comes from updating the **output layer $W^O$**, In our work, however, the linear decrease **comes from $W^V$ in attention head**. Our result provides a **different** insight from [1]: Optimization on the **attention head alone** can achieve linear convergence rate. In our analysis, the same linear convergence rate can be derived if we include $W^O$ in the optimization, but it remain unclear whether $W^V$, or even the attention head contributes to the linear decrease since updating $W^O$ alone can already lead to the same result. Thus, leaving out $W^O$ makes our insight **more clear** regarding the role of $W^Q,W^K,W^V$.

**Comment 2**: Comparison of assumptions in our Theorem 2 and [1]. New insight from the conditions.

**Response**:

**Comparison of conditions**: The conditions in [1] and our Theorem both consist of two parts: the network size and the scale of initial weights. We will compare the two conditions below:
(1) Network size: [1] considers the single head attention mechanism, which requires $D\geq n,d\geq N$. It means the embedding dimension and model size are lower bounded by sequence length, sample size, respectively. In our work, we require $Nn\leq HD$, while $d$ can be small. If we compute the total number of neurons in each variable of the attention mechanism in $H=1$ case, then the lower bound of total neuron number in our result is $\Omega(Nn)$, which is **exactly the same** as [1]! This result shows that our network size lower bound is **consistent** with the literature.
(2) Scale of initial weights: The initialization  conditions in two papers are very similar; it is **almost equally difficult** to satisfy the initialization conditions in two papers. The initialization condition in (8) implies that: 1. Both works require the smallest singular value of the attention head $B_0$ ($\underline{\lambda}^B$ in our paper) not to be too small; 2. Both works require the scale of each $W_h^Q, W_h^K$, and $W^V$ not to be too large; 3. Both works require the initial weight not to be too far from the global optimal solution; 4. Our Equation (8) requires the scale of $W^O$ to be large, while in Wu et al. [2024], the scale of $W^O$ has an **upper bound**. From above comparison, we can conclude that the initial conditions in two papers are **very similar**, except for the requirement for the scale of $W^O$, which is opposite. Further, we can verify that the gap between lower bounds for the smallest singular value of attention head ($\bar{\lambda}^B$ in our paper) is constant level, as well as the gap between the upper bounds for the scale of $W^Q,W^K,W^K$. The only difference is the scale of $W^O$, which results from the difference technique between papers: The linear decrease of loss function in [1] comes from updating $W^O$, while in our paper, the linear decrease of loss function in our paper comes from updating $W^V$. Please refer to the response to the previous question (**Difference in insight**).

**Regarding the new insight**: As we discussed in the previous point, the new insight we can derive from Theorem 2 is that: Optimization on the **attention head alone** can achieve a linear convergence rate. Please refer to the response to the previous question (**Difference in insight**).

[1] Wu, Yongtao, et al. "On the convergence of encoder-only shallow transformers." Advances in Neural Information Processing Systems 36 (2024).

---

### Decision · Program_Chairs · 2024-09-25

**Decision:**

Accept (poster)

**Comment:**

All reviewers agreed this paper should be accepted: it addresses an important problem, the analysis is insightful and novel, and the writing is clear. The author response addressed reviewer concerns causing a reviewer to increase their score and confidence. A clear accept. Authors: you've already indicated that you will update the submission to respond to reviewer changes, if you could double check their comments for any recommendation you may have missed on accident that would be great! The paper will make a great contribution to the conference!